# Research on Operation and Financing Strategy of an Emission-Dependent Supply Chain under Variable Transportation Fee Strategy

**Keyong Zhang [1], Chunxia Li [1,\*] and Jianming Yao [2]**

[1]   School of Economics and Management, North University of China, Taiyuan 030051, China
[2]   School of Business, Renmin University of China, Beijing 100872, China
\*   Correspondence: lichunxia960125@163.com; Tel.: +86-181-3510-4429

**Abstract:** Under the carbon cap-and-trade mechanism, we consider an emission-dependent supply chain consisting of a supplier, a manufacturer, and a 3PL firm that adopts variable transportation fee strategy. Five models on the basis of the supplier and manufacturer with or without capital constraints are considered to discuss members' optimal decisions. The insights are obtained as follows. First, the ordering quantity under 3PL financing service is larger than that under two firms are well-funded when the transportation fee or carbon emission is less than a certain constant. The variable transportation fee strategy and members' carbon emission reduction behavior are beneficial to each supply chain participant. Second, the carbon emissions of members decide whose capital constraint is more beneficial to 3PL firm, and 3PL prefers to cooperate with a medium rich manufacturer (rich supplier). Third, the external financing modes are analyzed to get the 3PL financing service can create new value for the manufacturer and 3PL if the transportation fee is below a threshold, and this threshold increases with the manufacturer's carbon emission. When the transportation fee is larger than a threshold, a capital-constrained supplier will choose bank financing, and this threshold decreases with the supplier's carbon emission. Finally, we demonstrate that the manufacturer's loss aversion (carbon cap) can increase (decrease) its bankruptcy threshold.

**Keywords:** cap-and-trade policy; capital constrained; variable transportation fee strategy; 3PL financing service; carbon emissions

## 1. Introduction

In recent years, the carbon emissions contribute a lot to global warming, and many countries seek various carbon regulations to reduce the operation-related carbon emissions. Carbon cap-and-trade system is more efficient in reducing the carbon emissions than other policies. Thus, this policy is widely adopted by many countries [1,2]. For example, the European Union implemented cap-and-trade policy in 2005, then the whole regions of Tokyo and California began testing this system [3]. The main objective of the Paris Agreement is to control the global average temperature increase in this century to less than 2 degrees Celsius [4]. Responding to this target, China, as the world's largest developing country, has issued its "National Plan on Climate Change". This plan put forward a goal that achieves reduction carbon emission by 40–50% in 2020 (relative to 2005 levels). In order to accomplish this task, China officially launched the national unified carbon emission trading market in 2017, which is the first developing country to control carbon emissions with cap-and-trade system. In this context, many enterprises have begun to market low-carbon processed products, such as Super Species and Freshhema [5].

However, the low-carbon supply chain involves many links and requires a lot of capital investments. Therefore, firms often face financial constraints in actual operation, especially the start-up and

small-medium-sized ones. In order to relieve this pressure, the capital-constrained firms choose financing from other members (internal financing) or financial institutions (external financing) [6,7]. For instance, Industrial Bank established cooperative relationship with six firms to provide carbon emission cap pledge loans for them in 2014. According to the data (website http://www.tanjiaoyi.com.), we know that about $31.8 million of loans were provided to companies with carbon emission cap as collateral in 2017. Therefore, some scholars try to incorporate capital situation of the emission-dependent firms into their operational decision-making [8,9].

Regarding the researches of 3PL firm, most scholars think that 3PL only provides logistics service for firms, which may include transportation and warehousing [10–12]. However, the increasing competition among 3PL firms leads to it must develop their own business scope. For example, the UPS and FedEx choose to offer financing service for their clients, which can create additional values for themselves. This is because the 3PL financing mode greatly reduces the firms' financing risks by coordinating the financial flow and material flow in the supply chain. As one of the main logistics in China, SINOTRANS has achieved cooperation with 10 banks to provide financing service for enterprises in 2007 [13]. Eternal Asia also provides both logistics and financing service for SMEs, which makes its revenues growth rate 6% higher than industry average rate [14]. However, even though the 3PL financing mode has been widely adopted in practice, there are few studies on this topic [13–15].

Based on the above discussions, we know that 3PL can acts two roles in practice and cap-and-trade regulation affect the operation and financing strategy of supply chain. Thus, we try to introduce the 3PL's transportation and financing service into an emission-dependent supply chain with cap-and-trade system. We also assume that 3PL offers a variable transportation fee strategy [15]. The above-mentioned findings raise questions as follows. How does the capital-constrained supplier and manufacturer make optimal operational decisions when 3PL provides financing service? What are the impacts of cap-and-trade system, 3PL's variable transportation fee, and members' initial capital on the operations and financing decisions of firms? How does the 3PL firm offers variable transportation fee and makes interest rates when it needs to act different roles? Does the 3PL's variable transportation fee strategy benefit supplier, manufacturer and 3PL firm? What the differences of members' decision-making and profits under different financing situations? When different financing modes are viable, what are the financing preferences of each enterprise?

To answer these questions, we consider a three-echelon supply chain consisting of a manufacturer, a supplier and a 3PL firm. The government enacts cap-and-trade policy in which the manufacturer and supplier can trade carbon caps in market. In this context, we design five models on the basis of the supplier and manufacturer with or without capital constraints, that is, two firms without capital constraints (*nn*), the supplier without capital constraint while the manufacturer with capital constraint (*st* and *lt*), the supplier with capital constraint while the manufacturer without capital constraint (*tl*), and the two firms with capital constraints (*tt*). Through the analysis and solutions of models, we get the optimal ordering quantity, wholesale price and loan interest rate in each case. We explore the impacts of cap-and-trade system, variable transportation fee strategy and manufacturer's loss aversion on financing operations and profits of the supply chain. And the selection of financing modes for supplier and manufacturer is also discussed. The results of this paper will help capital-constrained firms determine their strategies under carbon emission limits.

The remainder of the paper is organized as follows. Section 2 introduces the related literature. Section 3 describes the study problems and presents basic assumptions and notations. Section 4 designs models and demonstrates the optimal solutions in five models. Section 5 considers the external financing channels and impact of manufacturer's risk aversion. Section 6 presents the analysis of the optimal decisions. Section 7 provides several numerical examples to further illustrate and verify the results. Finally, Section 8 concludes the main insights and outlines directions for future research. All proofs are provided in Appendices A–C.

## 2. Literature Review

Considering environmental regulation, we design different financing models in a low-carbon situation, as well as analyze the transportation fee strategies. Therefore, our paper is mainly related to three streams of literature.

### 2.1. Cap-And-Trade System

With the increasing importance of sustainable economy, many countries have enacted a serious of carbon legislations, thus, there is growing interest in the effects of carbon policies on strategies of the supply chain [16–18]. Cheng et al. [19] explain how low-carbon policy impacts the interests of businesses, consumers, and policymakers, and they also emphasize the impacts of carbon labelling scheme on optimal strategies. Some researches discuss the comparison between cap-and-trade policy and other environment legislations. Zhang et al. [1] demonstrate that cap-and-trade policy is more efficient to reduce the carbon emissions than other policies. Drake et al. [2] compare the effects of cap-and-trade with that of carbon tax policy on the optimal decisions of enterprises, they propose that cap-and-trade system is easier to implement and creates greater expected profit.

Some researches explore the optimal decision-making of supply chain under the cap-and-trade system. For example, Yuan et al. [20] quantity the impacts of carbon trading price, members' emissions and carbon cap on the performances of firms with information asymmetry. Wang et al. [21] incorporate cap-and-trade system and consumer's low-carbon preference into dual-channel supply chain's operational decision-making, they find that members' profits increase with carbon price only when a product's green degree is high enough. There are some studies focus on the influence of cap-and-trade system on production, ordering quantity and inventory. Battini et al. [22] design a bi-objective lot-sizing model to analyze the manager's specific purchasing problem when carbon costs and emissions are kept separated. Under the cap-and-trade mechanism, Hua et al. [23] explore the firm's inventory management by using EOQ model and demonstrate how firms manage carbon footprints. Jaber et al. [24] consider a two-level supply chain model under the coordination mechanism and discuss different emission trading schemes and inventory strategies. García-Alvarado et al. [25] emphasize the importance of restructuring inventory policies by exploring the firm's inventory management.

As we can see, there has been a lot of studies on cap-and-trade system, but they do not consider the supply chain's financing problems and always assume that only one firm is imposed carbon regulation. With the intensification of competition, more and more firms (including upstream and downstream) are facing financial constraints in their operational practice. Therefore, it is very meaningful to explore the financing situations of an emission-dependent supply chain that manufacturer and supplier are regulated by cap-and-trade scheme.

### 2.2. Integrated Management of Operation and Finance

In recent years, supply chain financing has attracted more and more attention among operational practices and academic fields. According to different sources of capitals, supply chain financing channels can be divided into internal financing and external financing.

Some researches focus on capital-constrained firms apply for financing help from other members of the supply chain. These studies cover a variety of areas, such as operation, financing, and cooperation. For example, Chen et al. [6] explore the influence of trade credit and limited liability on optimal strategies, and they demonstrate that trade credit not only creates new value for firms but also achieves partly coordination of the supply chain. Rui et al. and Wang et al. [26,27] believe that trade credit can incite capital-constrained retailer to increase ordering quantity and make a partial payment for products. Lee et al. [28] propose that trade credit is more used by suppliers with smaller market share and it can as a competitive tool for suppliers with weak market power. Li et al. [29] combine the supplier's trade credit term with the buyer's order decision to make an optimal dynamic trade credit term decision. Other researches also focus on the influence of external financing on production

and inventory management. Jing et al. [7] investigate the optimal strategies between a bank and a capital-constrained retailer, the results show that unique equilibrium is bank credit if production cost is relatively high. Kouvelis et al. [30] assume that the supplier and retailer are facing financial constraints and apply for financing from bank, they also explore the influence of firm's bankruptcy costs. Tunca et al. [31] find that channel costs of the supply chain are decreased if retailer acts intermediation to help a capital-constrained supplier get financing from bank. On the basis of practical application, the differences between internal financing and external financing are examined. Chen, X.F., and Kouvelis et al. [32,33] assume that retailer may get financing from manufacturer or bank, and they find that trade credit integrates channels and performs better than bank financing. Yan et al. [34] compare bank with partial trade guarantee through a bilevel Stackelberg game model. The results show that partial trade guarantee can achieve profit maximization and channel coordination when guarantee coefficient is suitable. Li et al. [35] consider a capital-constrained retailer and a risk-averse supplier in two scenarios. The equilibrium solutions demonstrate that risk aversion degree and credit guarantee coefficient have significant effects on the selection of financing mode for retailer.

The above studies mainly analyze supply chain financing in a traditional supply chain. However, with the increasing importance of low-carbon economy, many scholars began to discuss supply chain financing in a low-carbon situation. For example, Deng et al. [8] explore the optimal carbon emission reduction mode selection strategy by comparing different operational decisions. Their results show that the retailer's initial capital has an important impact on the selection of carbon emission reduction mode. Sarkar et al. [9] incorporate the cost of carbon emissions into the environmental impact on total profits to design a synergic economic order quantity model. Based on a fuzzy multi-criteria evaluation method combined with Topsis, Liang et al. [36] propose a new SME financing evaluation model. Yang et al. [37] develop four models on the basis of two cooperation models to discuss the firm's carbon emission reduction decisions. They find that compare with other modes, the SCCF pattern can not only increase member's profits, but also achieve effectively control the total carbon emissions. Some studies also specifically explore how firms make their financing decisions under the cap-and-trade system. Cao et al. [38] assume that a capital-constrained retailer solves funding problem through trade credit, and obtain the retailer's optimal financial decisions under the cap-and-trade system. Qin et al. [39] investigate a capital-constrained manufacturer's carbon emission reductions under the cap-and-trade mechanism. In order to alleviate financial pressure, they also design two contracts: greening financing and cost sharing. Cao et al. [40] demonstrate that the existence of investments in carbon abatement has no impact on the selection of financing mode, and trade credit is a unique financing equilibrium for the manufacturers.

In the available researches, some studies analyze the applicability of cap-and-trade system, some studies explore the firms' decision-making under different financing modes, and others discuss the firms' financing decisions in a low-carbon situation. But they all ignore that the well-funded 3PL firm can also provide financing service for the capital-constrained members in order to expand business scope. This has become a new financing mode in internal financing.

*2.3. 3PL Financing Service*

According to the studies of Chen et al. [14], we know that 3PL firm can act in different roles in operational practice. The first is that 3PL firm only provides traditional transportation service to other partners, and the second is that 3PL firm provides both transportation and financing service to capital-constrained members.

We find that much research examines the first role of 3PL firm [10–12], but there are very few studies explore its second role (providing financing service). Chen et al. [13] analyze and compare the bank financing mode with the 3PL financing mode, and conclude that the overall profits under 3PL financing is higher than that under bank financing. Chen et al. [14] also find that 3PL firm obtain payment delay arrangements from the supplier can alleviate the high financing costs of a capital-constrained retailer and achieve the development of each firm. And by extending their model to include multiple retailers, they demonstrate that the supply chain's profit is higher under leadership

by the 3PL than that under leadership by the supplier. Zhou et al. [41] develop three models under manufacturer guarantor financing (MG) and 3PL guarantor financing (LG). The results show that the leader's economies of scale in operational cost decide the follower's financing preference. With the increasing development of 3PL firm, it is particularly important to discuss the different roles of 3PL in supply chain financing management.

The closest work to ours is perhaps that by Huang et al. [15], who consider that 3PL firm provides transportation and financing service for a capital-constrained retailer. The impact of 3PL's variable transportation fee strategy on the supply chain performance is discussed. they conclude that 3PL financing service is a feasible financing mode for the retailer. However, these authors believe that only downstream member has financial problem and upstream member is always well-funded. With the expansion of production scale, the supplier's initial capital can't meet the order demand of downstream member. This problem brings the risk of shortage to downstream member, thus affecting the stable development of the entire supply chain. And since the 3PL financing mode is adopted by many enterprises, thus, we introduce this mode into an emission-dependent supply chain under the cap-and-trade system to discuss how capital-constrained members make their optimal decisions. Our contributions can be concluded as follows: First, we consider an emission-dependent supply chain in which both the manufacturer and supplier are regulated by cap-and-trade policy. In this context, we develop five models on the basis of the supplier and manufacturer with or without financial constraints, which take full account of the supply chain funding situation. Second, we broke through the limitation of the traditional role of 3PL firm, namely, we assume that 3PL can provide transportation and financing service for capital-constrained members and it also can offer a variable transportation fee strategy. Third, by analyzing and comparing the optimal results, the comprehensive impacts of cap-and-trade system and variable transportation fee strategy on supply chain financing performance are demonstrated. Finally, the selection of financing mode for supplier and manufacturer is discussed when the internal and external financing modes are feasible. And we provide several numerical experiments to discuss the influencing factors of the financing mode selection.

## 3. Modeling Framework of Supply Chain Financing

### 3.1. Problem Description

In this paper, we investigate an emission-dependent supply chain with a supplier (denoted as s), a 3PL firm (denoted as l) and a manufacturer (denoted as m). To curb carbon emissions, the government enacts different carbon emission caps ($G, M$) on supplier and manufacturer. In the marketing period, the manufacturer orders $q$ from supplier at a unit price of $\omega$, then the 3PL ships the goods to the manufacturer. The manufacturer sells the products at another price $p$ at last. We consider that 3PL has a deep pocket and offers a variable transportation fee $t$. In order to develop business scope, 3PL also provides financing service for capital-constrained members based on loan interest rate $r$. Figure 1 illustrates the model structure of the supply chain system. Therefore, five models ($nn, st, lt, tl, tt$) are designed on the basis of whether the supplier and manufacturer are facing capital constraints.

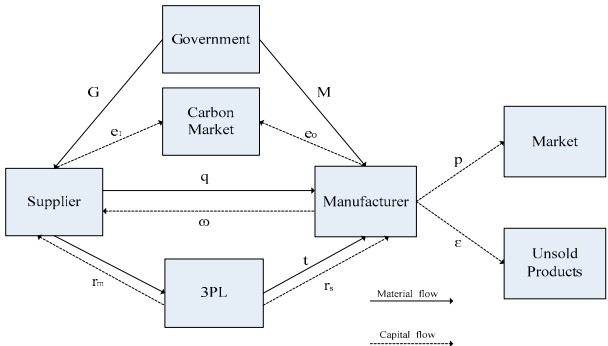

**Figure 1.** Model structure of the supply chain system.

### 3.2. Notation and Assumption

To guarantee the existence and uniqueness of solution to the optimization problem, we use the following assumptions (see [8,9,41]):

(1) Due to seasonal or other factors, we assume that market demand is uncertain (see [41]). The probability density function and cumulative distribution function of the demand distribution are $f(x)$ and $F(x)$, respectively, where $f(x) > 0$ on $[0, +\infty)$. The unique complementary and inverse function of CDF are $\overline{F}(x) = 1 - F(x)$ and $F^{-1}(x)$. We assume that $F(x)$ is twice differentiable, strictly increasing, and absolutely continuous. In addition, the hazard function and generalized failure rate are $h(x) = \frac{f(x)}{\overline{F}(x)}$ and $H(x) = xh(x)$, respectively. The demand distribution has an increasing failure rate (IFR), namely, $h(x)$ and $H(x)$ are monotonically increasing function of market demand $x$.

(2) The supplier, 3PL firm and manufacturer have the same belief about each other's situation and market demand, and all of them are pursue the maximization of profits.

(3) Under the carbon-and-trade policy, supplier and manufacturer initially obtain a certain number of carbon emission caps over a planning horizon from the government, if carbon emissions exceed (below) the carbon cap, they can sell (buy) carbon emission cap in the carbon market at a trading price $p_c$.

(4) The selling season is short, for simplicity, we assume that risk-free interest rate and time value are zero. In addition, the salvage value of the unit unsold product in the end of the selling season is $\varepsilon$.

(5) Without loss of generality, we can get the inequalities $n < t, \varepsilon < c < \omega, c(1+r) < \omega, (\omega + t)(1+r) < p$.

For convenience, we use the notations in Table 1. throughout the paper.

**Table 1.** Notations for parameters and variables.

| | |
|---|---|
| **Decision Variables** | |
| $\omega_j$ | Wholesale price, $j = nn, st, lt, tl, tt$ |
| $q_j$ | Ordering quantity, $j = nn, st, lt, tl, tt$ |
| $r_j$ | Loan interest rate, $j = nn, st, lt, tl, tt$ |
| **Model Parameters** | |
| $c$ | Supplier's unit production cost |
| $G$ | Supplier's carbon caps |
| $M$ | Manufacturer's carbon caps |
| $p_c$ | Carbon trading price per unit product |
| $e_1$ | Carbon emission per unit product of the supplier |
| $e_0$ | Carbon emission per unit product of the manufacturer |
| $t_j$ | 3PL's unit transportation fee, $j = nn, st, lt, tl, tt$ |
| $n$ | 3PL's unit transportation cost |
| $p$ | Manufacturer's retail price |
| $\varepsilon$ | Salvage value of unit unsold product |
| $s$ | Supplier's initial capital |
| $b$ | Manufacturer's initial capital |
| $x$ | Actual sales of the manufacturer |
| $\lambda$ | Manufacturer's loss aversion |
| $\Pi_j^i$ | Profit of the supply chain member, $i = s, l, m, \ j = nn, st, lt, tl, tt$ |

The subscripts $j = nn, st, lt, tl, tt$ denotes the five scenarios respectively, namely, two firms without capital constraints (*nn*), the manufacturer with capital constraint (supplier financing service mode *st* and 3PL financing service mode *lt*), the supplier with capital constraint (*tl*), and the two firms with capital constraints (*tt*). The superscripts $i = s, l, m$ denotes the supplier, 3PL firm and manufacturer respectively. In the extension section, the subscripts *bt*, *tb* denotes the manufacturer and supplier apply

for financing from bank respectively, the subscripts *sa*, *la* denotes the manufacturer is loss-averse under supplier financing mode and 3PL financing mode respectively.

## 4. The Equilibrium of Supply Chain System under Different Financing Modes

### 4.1. The Equilibrium under Two Firms Are without Capital Constraints

In this section, we investigate how supplier and manufacturer make decisions to maximize their profits when they are well-funded. The 3PL firm only needs to offer traditional transportation service. Therefore, we establish a Stackelberg game model with supplier as the leader and manufacturer as the follower. Figure 2 illustrates the sequences of events and decisions (*nn*).

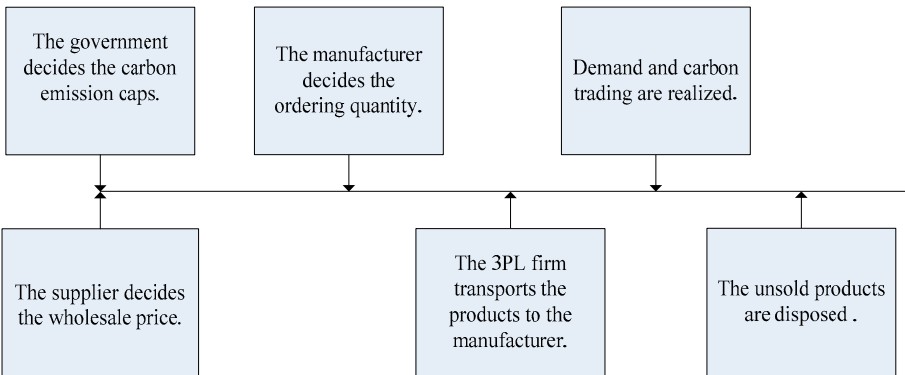

**Figure 2.** Sequences of events and decisions (*nn*).

Under the cap-and-trade mechanism, a well-funded manufacturer determines the ordering quantity to maximize its profit. Therefore, the manufacturer's decision objective under *nn* can be formulated as follows:

$$\Pi_{nn}^m(q_{nn}) = \max_{q_{nn}} E[p\min(q_{nn}, x) + \varepsilon(q_{nn} - x)^+ + p_c(M - e_0 q_{nn}) - (\omega_{nn} + t_{nn})q_{nn}] \tag{1}$$

We analyze the supplier's decision on setting the wholesale price to maximize its profit. Therefore, the well-funded supplier's optimization problem is as below:

$$\Pi_{nn}^s(\omega_{nn}) = \max_{\omega_{nn}} E[(\omega_{nn} - c)q_{nn}^*(\omega_{nn}) + p_c(G - e_1 q_{nn}^*(\omega_{nn}))] \tag{2}$$

Backward induction is adopted to solve the supply chain optimization problem. Thus, we can form our first proposition (see Appendix A).

### 4.2. The Equilibrium under Manufacturer with Capital Constraint

In this section, the manufacturer faces capital constraint, which means $(\omega_{st} + t_{st})q_{st} > b$. The manufacturer tends to borrow capital from other partners of the supply chain when it can't get loans from external institutions. In this context, we assume that manufacturer may adopt two financing methods: loaning from a supplier or loaning from a 3PL firm.

#### 4.2.1. The Optimal Strategies under Supplier Financing Service Mode

In this subsection, a capital-constrained manufacturer apply for financing help from well-funded supplier. The supplier as the Stackelberg leader to decide the wholesale price $\omega_{st}$ and loan interest rate $r_{st}$. According to the supplier's decisions, the manufacturer determines the ordering quantity $q_{st}$. Figure 3 illustrates the sequences of events and decisions (*st*).

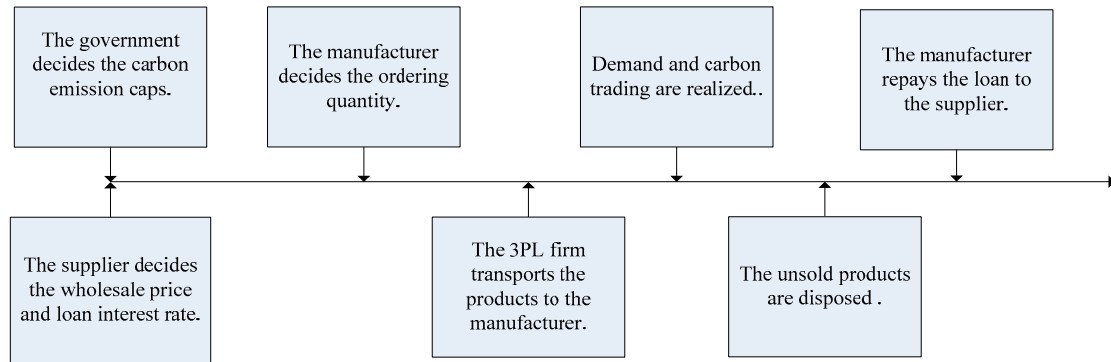

**Figure 3.** Sequences of events and decisions (*st*).

Under the supplier financing mode, the manufacturer needs to repay the corresponding principal and interest $((\omega_{st} + t_{st})q_{st} - b)(1 + r_{st})$ to supplier at the end of the sales period. Therefore, the manufacturer's decision objective is formulated as follows:

$$\Pi_{st}^m(q_{st}) = \max_{q_{st}} E[p\min(q_{st}, x) + \varepsilon(q_{st} - x)^+ + p_c(M - e_0 q_{st}) - ((\omega_{st} + t_{st})q_{st} - b)(1 + r_{st}) - b] \quad (3)$$

The supplier set $\omega_{st}^*$ and $r_{st}^*$ to ensure profit maximization, thus, its decision objective is as:

$$\Pi_{st}^s(\omega_{st}) = \max_{\omega_{st}, r_{st}} E[(\omega_{st} - c)q_{st}^*(\omega_{st}) + p_c(G - e_1 q_{st}^*(\omega_{st})) + ((\omega_{st} + t_{st})q_{st}^*(\omega_{st}) - b)r_{st}] \quad (4)$$

Proposition 2 can be formed by adopting backward induction to solve the supply chain optimization problem (see Appendix A).

### 4.2.2. The Optimal Strategies under 3PL Firm Financing Service Mode

Consider the situation that a manufacturer's capital deficit can be financed by 3PL firm. In this framework, the supplier as leader to give a wholesale price $\omega_{lt}$ preferentially. Then the 3PL firm acts two roles to determine the loan interest rate $r_{lt}$ and transportation fee strategy. Based on the above two variables, the capital-constrained manufacturer determines the ordering quantity $q_{lt}$. Figure 4 illustrates the sequences of events and decisions (*lt*).

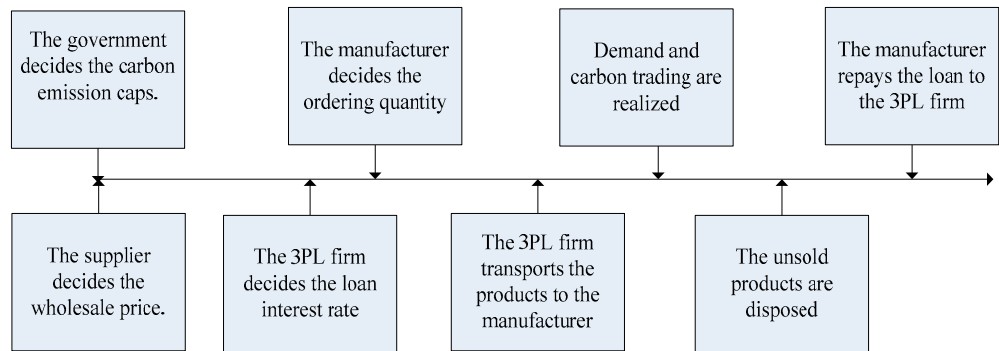

**Figure 4.** Sequences of events and decisions (*lt*).

Under the cap-and-trade mechanism, the capital-constrained manufacturer needs to repay $((\omega_{lt} + t_{lt})q_{lt} - b)(1 + r_{lt})$ to 3PL firm at the end of the sales period, thus, the manufacturer's decision objective can be presented as follows:

$$\Pi_{lt}^m(q_{lt}) = \max_{q_{lt}} E[p\min(q_{lt}, x) + \varepsilon(q_{lt} - x)^+ + p_c(M - e_0 q_{lt}) - ((\omega_{lt} + t_{lt})q_{lt} - b)(1 + r_{lt}) - b] \quad (5)$$

The 3PL can obtain revenue $((\omega_{lt} + t_{lt})q_{lt}^*(r_{lt}) - b)r_{lt}$ from the manufacturer and yields transportation profit as $(t_{lt} - n)q_{lt}^*(r_{lt})$. Thus, we define the 3PL's profit optimization formula as:

$$\Pi_{lt}^l(\omega_{lt}) = \max_{r_{lt}} E[((\omega_{lt} + t_{lt})q_{lt}^*(r_{lt}) - b)r_{lt} + (t_{lt} - n)q_{lt}^*(r_{lt})] \tag{6}$$

Under the 3PL financing service mode, the supplier's optimization problem is as bellow:

$$\Pi_{lt}^s(\omega_{lt}) = \max_{\omega_{lt}} E[(\omega_{lt} - c)q_{lt}^*(\omega_{lt}) + p_c(G - e_1 q_{lt}^*(\omega_{lt}))] \tag{7}$$

Proposition 3 can be formed by adopting backward induction to solve the supply chain optimization problem (see Appendix A).

### 4.3. The Equilibrium under Supplier with Capital Constraint

In this section, we consider that the supplier faces capital constraint, which means $cq_{tl} > s$. First, the 3PL lends a capital of $cq_{tl} - s$ to supplier at interest rate $r_{tl}$. Then the supplier as a sub-leader to decide the wholesale price $\omega_{tl}$. Finally, the manufacturer performed as follows to determine the ordering quantity $q_{tl}$. Figure 5 illustrates the sequences of events and decisions (*tl*).

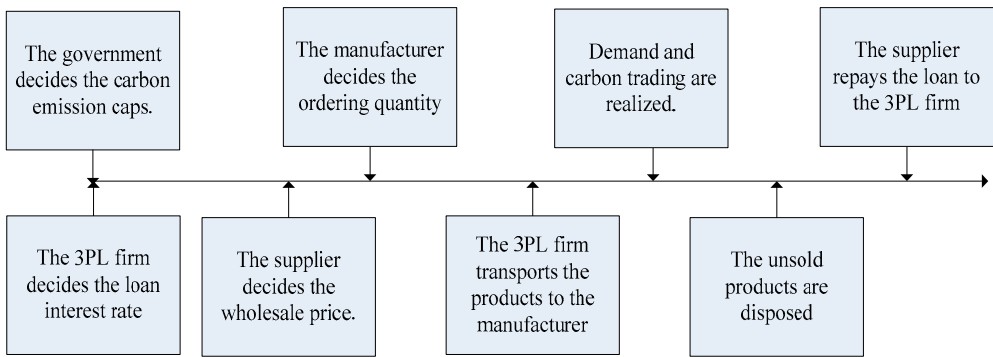

**Figure 5.** Sequences of events and decisions (*tl*).

In this context, the manufacturer's decision objective is formulated as follows:

$$\Pi_{tl}^m(q_{tl}) = \max_{q_{tl}} E[p\min(q_{tl}, x) + \varepsilon(q_{tl} - x)^+ + p_c(M - e_0 q_{tl}) - (\omega_{tl} + t_{tl})q_{tl}] \tag{8}$$

At the end of the sales horizon, the supplier needs to repay $(cq_{tl}^*(\omega_{tl}) - s)(1 + r_{tl})$ to the 3PL. Thus, the supplier's decision objective can be presented as bellow:

$$\Pi_{tl}^s(\omega_{tl}) = \max_{\omega_{tl}} E[\omega_{tl}q_{tl}^*(\omega_{tl}) + p_c(G - e_1 q_{tl}^*(\omega_{tl})) - (cq_{tl}^*(\omega_{tl}) - s)(1 + r_{tl}) - s] \tag{9}$$

Since the 3PL firm acts two roles in the operation process, it can yield the revenue as $(cq_{tl}^*(\omega_{tl}) - s)r_{tl}$ and transportation profit $(t_{tl} - n)q_{tl}^*(r_{tl})$. Thus, the 3PL's problem is written as:

$$\Pi_{tl}^l(r_{tl}) = \max_{r_{tl}} E[(cq_{tl}^*(r_{tl}) - b)r_{tl} + (t_{tl} - n)q_{tl}^*(r_{tl})] \tag{10}$$

Proposition 4 can be formed by adopting backward induction to solve the supply chain optimization problem (see Appendix A).

### 4.4. The Equilibrium under Two Firms Are with Capital Constraints

#### 4.4.1. The Optimal Strategies of Supplier and Manufacturer

In this section, we explore a situation that the supplier and manufacturer are face capital constraints, which means $cq_{tt} > s$ and $(\omega_{tt} + t_{tt})q_{tt} > b$. At the beginning, the 3PL firm acts main leader to set the loan interest rate $r_s$ and $r_m$. Then, the supplier acts sub-leader to decide the wholesale price $\omega_{tt}$. Based on the above variables, the manufacturer serves as a follower to determine the ordering quantity $q_{tt}$. Figure 6 illustrates the sequences of events and decisions (*tt*).

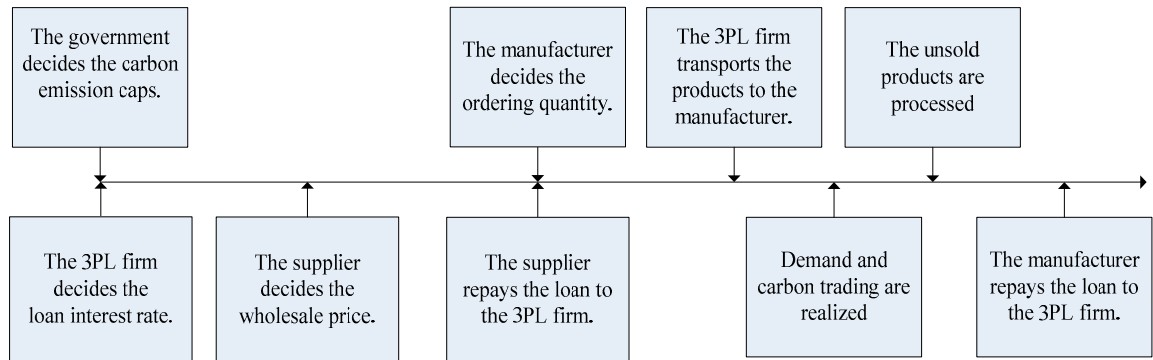

**Figure 6.** Sequences of events and decisions (*tt*).

According to the above analysis, the optimization problem of manufacturer and supplier are exclusively written as follows:

$$\Pi_{tt}^m(q_{tt}) = \max_{q_{tt}} E[p\min(q_{tt}, x) + \varepsilon(q_{tt} - x)^+ + p_c(M - e_0 q_{tt}) - ((\omega_{tt} + t_{tt})q_{tt} - b)(1 + r_{tt}) - b] \quad (11)$$

$$\Pi_{tt}^s(\omega_{tt}) = \max_{\omega_{tt}} E[\omega_{tt} q_{tt}^*(\omega_{tt}) + p_c(G - e_1 q_{tt}^*(\omega_{tt})) - (cq_{tt}^*(\omega_{tt}) - s)(1 + r_s) - s] \quad (12)$$

#### 4.4.2. The Optimal Strategies of 3PL Firm

The 3PL firm acts as a capital pool to finance supplier and manufacturer, while provides transportation service for the manufacturer. Therefore, the 3PL's payoff is made up of two components, that is, transportation and financing revenues. The 3PL's profit-objective can be written as:

$$\begin{aligned} \max \quad & \Pi_{tt}^l(r_s, r_m) = \Pi_{ts}^l + \Pi_{tm}^l \\ s.t. \quad & \Pi_{ts}^l = E[(cq_{tt}^*(r_s) - s)r_s] \\ & \Pi_{tm}^l = E[((\omega_{tt} + t_{tt})q_{tt}^*(r_m) - b)r_m + (t_{tt} - n)q_{tt}^*(r_m)] \end{aligned} \quad (13)$$

The optimal loan interest rate decisions are carried out separately. The loan interest rate to the manufacturer is analyzed first, the 3PL's expected profits from manufacturer and supplier can be written as follows:

$$\Pi_{tm}^l(r_m) = \max_{r_m} E[((\omega_{tt} + t_{tt})q_{tt}^*(r_m) - b)r_m + (t_{tt} - n)q_{tt}^*(r_m)] \quad (14)$$

$$\Pi_{ts}^l = \max_{r_s} E[(cq_{tt}^*(r_s) - s)r_s] \quad (15)$$

Backward induction is adopted to solve the supply chain optimization problem. Thus, we can obtain Proposition 5 (see Appendix A).

## 5. Extension

### 5.1. The Equilibrium under Bank Financing

In this section, we assume that the supply chain members' capital deficits can be financed by bank loans, and the selection of financing mode for supplier and manufacturer is discussed.

#### 5.1.1. The Optimal Strategies under Manufacturer with Capital Constraint

Under the cap-and-trade mechanism, the supplier offers a wholesale price $\omega_{bt}$. Then the bank determines the loan interest rate $r_{bt}$ and lends $(\omega_{bt} + t_{bt})q_{bt} - b$ to capital-constrained manufacturer. Finally, the manufacturer determines the ordering quantity $q_{bt}$. Thus, the optimization problems can be expressed as follows:

$$\Pi_{bt}^m(q_{bt}) = \max_{q_{bt}} E[p\min(q_{bt}, x) + \varepsilon(q_{bt} - x)^+ + p_c(M - e_0 q_{bt}) - ((\omega_{bt} + t_{bt})q_{bt} - b)(1 + r_{bt}) - b] \quad (16)$$

$$\Pi_{bt}^b(r_{bt}) = \max_{r_{bt}} E[(\omega_{bt} + t_{bt})q_{bt}^*(r_{bt}) - b)r_{bt}] \quad (17)$$

$$\Pi_{bt}^s(\omega_{bt}) = \max_{\omega_{bt}} E[(\omega_{bt} - c)q_{bt}^*(\omega_{bt}) + p_c(G - e_1 q_{bt}^*(\omega_{bt}))] \quad (18)$$

Solving the model through backward induction, we can obtain the optimal wholesale price as $\omega_{bt}^* = \frac{(1 + r_{bt}^*(\omega_{bt}^*))(H(q_{bt}^*(\omega_{bt}^*))t_{bt} + c + p_c e_1) + (p_c e_0 - \varepsilon)H(q_{bt}^*(\omega_{bt}^*))}{(1 - H(q_{bt}^*(\omega_{bt}^*)))(1 + r_{bt}^*(\omega_{bt}^*))}$, the optimal interest rate as $r_{bt}^* = \frac{((\omega_{bt} + t_{bt})q_{bt}^*(r_{bt}) - b)(p - \varepsilon)f(q_{bt}^*(r_{bt}))}{(\omega_{bt} + t_{bt})^2}$, and the optimal ordering quantity as $q_{bt}^* = \overline{F}^{-1}(\frac{(\omega_{bt} + t_{bt})(1 + r_{bt}) + p_c e_0 - \varepsilon}{p - \varepsilon})$.

#### 5.1.2. The Optimal Strategies under Supplier with Capital Constraint

We consider that a capital-constrained supplier borrows a capital of $cq_{tb} - s$ from the bank at interest rate $r_{tb}$ and determines the wholesale price $\omega_{tb}$. And the manufacturer decides the ordering quantity $q_{tb}$. Thus, the optimization problems can be expressed as follows:

$$\Pi_{tb}^m(q_{tb}) = \max_{q_{tb}} E[p\min(q_{tb}, x) + \varepsilon(q_{tb} - x)^+ + p_c(M - e_0 q_{tb}) - (\omega_{tb} + t_{tb})q_{tb}] \quad (19)$$

$$\Pi_{tb}^s(\omega_{tb}) = \max_{\omega_{tb}} E[\omega_{tb} q_{tb}^*(\omega_{tb}) + p_c(G - e_1 q_{tb}^*(\omega_{tb})) - (cq_{tb}^*(\omega_{tb}) - s)(1 + r_{tb}) - s] \quad (20)$$

$$\Pi_{tb}^b(r_{tb}) = \max_{r_{tb}} E[(cq_{tb}^* - b)r_{tb}] \quad (21)$$

Solving the model through backward induction, we can obtain the optimal interest rate as $r_{tb}^* = \frac{-f(q_{tb}^*(\omega_{tb}))(p - \varepsilon)(s - cq_{tb}^*(\omega_{tb}))}{c^2}$, the optimal wholesale price as $\omega_{tb}^* = \frac{H(q_{tb}^*(\omega_{tb}^*))(t_{tb} + p_c e_0 - \varepsilon) + p_c e_1 + c(1 + r_{tb})}{1 - H(q_{tb}^*(\omega_{tb}^*))}$, and the optimal ordering quantity as $q_{tb}^* = \overline{F}^{-1}(\frac{\omega_{tb} + t_{tb} + p_c e_0 - \varepsilon}{p - \varepsilon})$.

### 5.2. The Impact of Manufacturer's Loss Aversion

Due to the uncertainty of demand, the manufacturer may incur default risk and face a loss in financing schemes. Therefore, we consider a case that the manufacturer is loss-averse. Similar to the work of [42], we use the loss aversion utility function to measure the manufacturer's loss aversion.

#### 5.2.1. The Optimal Strategies under Supplier Financing Service Mode

Under the supplier financing service mode, we investigate a loss-averse and capital-constrained manufacturer. The optimization problems can be expressed as follows:

$$\Pi_{sa}^m(q_{sa}) = \max_{q_{sa}} E[p\min(q_{sa}, x) + \varepsilon(q_{sa} - x)^+ + p_c(M - e_0 q_{sa}) - ((\omega_{sa} + t_{sa})q_{sa} - b)(1 + r_{sa}) - b] \quad (22)$$

$$\Pi_{sa}^{s}(\omega_{sa}) = \max_{\omega_{sa}, r_{sa}} E[\min[p\min(q_{sa}^{*}(\omega_{sa}), x) + \varepsilon(q_{sa}^{*}(\omega_{sa}) - x)^{+} + p_{c}(M - e_{0}q_{sa}^{*}(\omega_{sa})) - b, ((\omega_{sa} + t_{sa})q_{sa}^{*}(\omega_{sa}) - b)(1 + r_{sa})]$$
$$-((\omega_{sa} + t_{sa})q_{sa}^{*}(\omega_{sa}) - b) + (\omega_{sa} - c)q_{sa}^{*}(\omega_{sa}) + p_{c}(G - e_{1}q_{sa}^{*}(\omega_{sa}))] \tag{23}$$

We identify $x_0 = \frac{((\omega_{sa}+t_{sa})q_{sa}-b)(1+r_{sa})+b-p_c(M-e_0q_{sa})-\varepsilon q_{sa}}{p-\varepsilon}$ is the bankruptcy threshold for capital-constrained manufacturer. Therefore, the loss-averse and capital-constrained manufacturer's expected utility can be written as:

$$U(\Pi_{sa}^{m}(q_{sa})) = \begin{cases} \lambda \int_{0}^{x_0} [px + \varepsilon(q_{sa} - x)^{+} + p_c(M - e_0 q_{sa}) - ((\omega_{sa} + t_{sa})q_{sa} - b)(1 + r_{sa}) - b]f(x)dx & if\ x < x_0 \\ \int_{x_0}^{q_{sa}} [px + \varepsilon(q_{sa} - x)^{+} + p_c(M - e_0 q_{sa}) - ((\omega_{sa} + t_{sa})q_{sa} - b)(1 + r_{sa}) - b]f(x)dx & if\ x_0 < x < q_{sa} \\ \int_{q_{la}}^{+\infty} [pq_{sa} + p_c(M - e_0 q_{sa}) - ((\omega_{sa} + t_{sa})q_{sa} - b)(1 + r_{sa}) - b]f(x)dx & if\ x > q_{sa} \end{cases} \tag{24}$$

Solving the model through backward induction, we can obtain Proposition 6 (see Appendix A).

### 5.2.2. The Optimal Strategies under 3PL Firm Financing Service Mode

In this subsection, we assume that 3PL offers the financing service for a loss-averse and capital-constrained manufacturer. The optimization problems can be written as follows:

$$\Pi_{la}^{m}(q_{la}) = \max_{q_{la}} E[p\min(q_{la}, x) + \varepsilon(q_{la} - x)^{+} + p_c(M - e_0 q_{la}) - ((\omega_{la} + t_{la})q_{la} - b)(1 + r_{la}) - b] \tag{25}$$

$$\Pi_{la}^{l}(r_{la}) = \max_{r_{la}} E[\min[p\min(q_{la}^{*}(r_{la}), x) + \varepsilon(q_{la}^{*}(r_{la}) - x)^{+} + p_c(M - e_0 q_{la}^{*}(r_{la})) - b, ((\omega_{la} + t_{la})q_{la}^{*}(r_{la}) - b)(1 + r_{la})]$$
$$-((\omega_{la} + t_{la})q_{la}^{*}(r_{la}) - b) + (t_{la} - n)q_{la}^{*}(r_{la})] \tag{26}$$

$$\Pi_{la}^{s}(\omega_{la}) = \max_{\omega_{la}} E[(\omega_{la} - c)q_{la}^{*}(\omega_{la}) + p_c(G - e_1 q_{la}^{*}(\omega_{la}))] \tag{27}$$

Solving the model through backward induction, we can obtain Proposition 7 (see Appendix A).

## 6. Results and Analysis

This section analyzes the relationship between some parameters and decision variables, as well as the comparison of optimal decisions among different situations. Through the following corollaries, we investigate the impacts of cap-and-trade system and variable transportation fee strategy on the optimal decision-making and profits of supply chain members.

**Corollary 1.** *The impacts of $\omega$, $r$, $t$, $p_c$, $e_0$, $e_1$ and $\varepsilon$ on the optimal ordering quantity $q^*$ are given as follows:*

(1)　*In all situations, $q_j^*$ decreases with $\omega_j$, $r_j$, $t_j$ and also decreases with $p_c$, $e_0$, $e_1$.*

(2)　*In all situations, $q_j^*$ increases with $\varepsilon$, where, $j = nn, st, lt, tl, tt$.*

Under the cap-and-trade mechanism, there is no doubt that the ordering quantity decreases with the supplier's wholesale price. Corollary 1(1) shows that the optimal ordering quantity decreases with the 3PL's interest rate to the manufacturer or supplier. As 3PL's interest rate to the supplier or manufacturer increases, supplier and manufacturer need to undertake additional interest rate charges, which leads to a reduction of the supplier's output and manufacturer's ordering quantity. The 3PL's unit transportation fee also increases the burden on manufacturer, thus, the manufacturer reduces its ordering quantity to protect profit and decrease loss risk. It is obvious to conclude that the supplier's output decreases with the cost per unit product. As shown in Corollary 1(1), with the increasing of carbon trading price and carbon emissions of supplier or manufacturer, more cost will be paid in the carbon market, which results in a reduction in production volume and ordering quantity. In addition, Corollary 1(2) indicates that a higher salvage value of unsold product can greatly decrease the manufacturer's loss risk, so manufacturer will order more products.

Corollary 1 indicates that increased the carbon trading price and carbon emissions of supplier or manufacturer (salvage value of unsold product) have negative (positive) effects on ordering quantity. And the 3PL's interest rate and transportation fee negatively affect the ordering quantity.

**Corollary 2.** *The impacts of* $r$, $p_c$, $e_0$, $e_1$ *and* $\varepsilon$ *on the optimal wholesale price* $\omega^*$ *are given as follows:*

(1)   *The optimal wholesale price* $\omega_{lt}^*$ *decreases with* $r_{lt}$, $\omega_{tl}^*$ *increases with* $r_{tl}$, $\omega_{tt}^*$ *decreases with* $r_m$ *and increases with* $r_s$.

(2)   *In all situations,* $\omega_j^*$ *increases with* $p_c$, $e_0$, $e_1$ *and decreases with* $\varepsilon$.

Corollary 2(1) shows that the supplier's optimal wholesale price increases with the 3PL's interest rate to the supplier and decreases with the 3PL's interest rate to the manufacturer. This is because the supplier needs to set a higher wholesale price to transfer huge costs when 3PL enacts a higher interest rate. In addition, a higher interest rate costs will cause a capital-constrained manufacturer to lose motivation to order products. Thus, the supplier has to lower wholesale price to stimulate the increase of ordering quantity, so that the scale effect can be achieved. Corollary 2(2) indicates that under the cap-and-trade mechanism, with the increasing of carbon trading price or carbon emissions (including the productive process and sales process), supplier needs pay more money in the carbon trading market, which leads to an increase in wholesale price. We also conclude that the higher salvage of unsold product can stimulate ordering quantity, thus, the supplier is motivated to adopt a conservative wholesale price strategy for more orders.

Corollary 2 indicates that as the carbon trading price or carbon emission increases, supplier should raise the wholesale price. The 3PL's interest rate to the manufacturer (supplier) has a negative (positive) effect on the wholesale price.

**Corollary 3.** *The impacts of* $\omega$, $t$, $p$, $\varepsilon$, $b$ *and* $s$ *on the optimal interest rate* $r^*$ *are given as follows:*

(1)   *In situation lt and tt,* $r_{lt}^*$ *decreases with* $\omega_{lt}$, $t_{lt}$; $r_m^*$ *decreases with* $\omega_{tt}$, $t_{tt}$.

(2)   *In situation tl and tt,* $r_{tl}^*$ *decreases with* $t_{tl}$; $r_s^*$ *decreases with* $t_{tt}$.

(3)   *In all situation,* $r_j^*$ *increases with* $p$ *and decreases with* $\varepsilon$.

(4)   *In situation lt, tl and tt,* $r_{lt}^*$ *and* $r_m^*$ *decrease with* $b$; $r_{tl}^*$ *and* $r_s^*$ *decrease with* $s$.

Corollary 3(1) shows that the 3PL's interest rate to the manufacturer decreases as the wholesale price increases. Intuitively, wholesale price is a cost to the manufacturer, thus, the 3PL will decrease its interest rate in order to improve the manufacturer's financing motivation. Corollary 3(1) and (2) indicate that the 3PL's interest rate to the manufacturer (supplier) decreases with the unit transportation fee. As the 3PL's transportation fee increases, the manufacturer needs to bear higher transportation charges and loss risk, which leads to a reduction in ordering quantity. In this context, both the supplier and manufacturer are lost the incentives to apply for financing from 3PL because their profits are damaged. Therefore, in order to develop financing business, the 3PL needs to reduce its loan interest rate to entice capital-constrained members to financing. On the contrary, Corollary 3(3) shows that the 3PL's interest rate to the manufacturer or supplier increases (decreases) with the retail price of product (salvage value of unsold product). The 3PL needs to increase (decrease) its interest rate to protect profit (stimulate ordering quantity) when retail price (salvage value) is high (low), which can improve the supply chain performance. Corollary 3(4) indicates that the 3PL's interest rate to the manufacturer (supplier) decreases with the manufacturer's (supplier's) initial capital. The reason is that the higher the initial capital, the lower loss risk faced by members. In this context, the 3PL firm can obtain new value by decreasing the loan interest rate.

Corollary 3 implies that as the transportation fee or salvage value of unsold product increases, the 3PL firm should decrease loan interest rate to the manufacturer and supplier. In addition, the manufacturer's (supplier's) initial capital has a negative effect on 3PL's interest rate to the manufacturer (supplier).

**Corollary 4.** *If the 3PL firm adopts fixed transportation fee strategy, that is,* $t_j = t$, *the relationship of optimal strategies under different models are given as follows:*

(1)   Under different models, the relationship of optimal ordering quantity satisfies $q_{st}^* = q_{nn}^* \geq q_{j6}^*$, where, $j6 = lt, tl, tt$.

(2)   Under different models, the relationship of optimal wholesale price satisfies $\omega_{lt}^* \leq \omega_{st}^* = \omega_{nn}^*$, if $p \leq t + p_c e_0 - \frac{H(q_{nn}^*)(t + p_c e_0 - e)}{(1 - H(q_{nn}^*))f(q_{tl}^*)} + \frac{H(q_{tl}^*)(t + p_c e_0 - e)}{(1 - H(q_{tl}^*))f(q_{tl}^*)}$, we have $\omega_{tl}^* \geq \omega_{st}^* = \omega_{nn}^*$, otherwise, $\omega_{tl}^* \leq \omega_{st}^* = \omega_{nn}^*$.

(3)   Under different models, the relationship of manufacturer's profit satisfies $\Pi_{j6}^m(q_{j6}^*) \leq \Pi_{j6max}^m(q_{j6}^*) \leq \Pi_{nn}^m(q_{nn}^*) = \Pi_{st}^m(q_{st}^*)$.

(4)   In situation nn and st, the relationship of supplier's profit satisfies $\Pi_{nn}^s(q_{nn}^*) = \Pi_{st}^s(q_{st}^*)$.

Corollary 4(1) shows that the optimal ordering quantity under model *st* is equal to that under model *nn*, and larger than that under model *lt*, *tl* and *tt*. This phenomenon indicates that the transaction costs of manufacturer and supplier will increase when 3PL firm adopts a fixed transportation fee strategy. This also proves the importance of variable transportation fee strategy in supply chain practice. And as Corollary 4(3) presents, the manufacturer's profit under model *nn* and *st* are larger than that under model *lt*, *tl* and *tt*. This result in the profits of overall supply chain under model *lt*, *tl* and *tt* are damaged. Intuitively, the supplier will set a lower wholesale price to entice capital-constrained manufacturer to order more products, so as shown in Corollary 4(2), the optimal wholesale price under model *lt* is less than that under model *nn* and *st*. And we can find that the optimal wholesale price under mode *tl* is larger (less) than that under model *nn* and *st* when retail price is less (larger) than a threshold value. If the retail price is less than a threshold, the supplier's extra cost from 3PL financing is relatively large, thus, the capital-constrained supplier will transfer this cost to the manufacturer. Otherwise, the capital-constrained supplier under model *tl* will decrease wholesale price to stimulate ordering quantity. Corollary 4(3) and (4) shows that the optimal profit of the manufacturer (supplier) under model *st* is equal to that under model *nn*, which means that capital-constrained manufacturer under model *st* does not need to bear additional costs, namely, supplier financing mode can coordinate the supply chain when manufacturer with financial constraint.

## 7. Numerical Analysis

In this section, we design several numerical examples to illustrate our main results and obtain some managerial insights. We assume that market demand $x$ as being uniformly distributed on the interval from 0 to 100. Other parameters are set up as follows: $p = 50$, $c = 10$, $G = 150$, $M = 100$, $e_1 = 1$, $e_0 = 0.5$, $p_c = 3$, $t_j = 12.5$, $n = 5$, $b = 200$, $s = 50$, $\varepsilon = 0$.

### 7.1. Impact of Relevant Parameters on Optimal Operational Strategies

As shown in Figure 7, we investigate the impacts of 3PL's unit transportation fee $t$ and carbon trading price $p_c$ on the optimal operational strategies of supply chain members. From Figure 7a, we find that $t$ is a cost for the manufacturer, which leads to a reduction in ordering quantity $q^*$. And $q^*$ under model *lt* is more than that under model *nn*, *st* and *tl* when $t$ is less than 10.5. This demonstrate that under 3PL financing service, manufacturer with capital constraint is better for $q^*$ than supplier with capital constraint when $t$ is less than 14.3. From Figure 7b, we observe that wholesale price $\omega^*$ decreases with $t$, and $\omega^*$ under model *tl* is larger than that under model *tt* if $t$ is lower than 13.5. It is intuitive that $t$ has a negative effect on manufacturer, thus, the supplier sets a lower (higher) $\omega^*$ under model *lt* (*tl*) to stimulate ordering quantity (protect interests), thus, $\omega^*$ under model *lt* is the lowest. Figure 7d,e illustrates that $q^*$ ($\omega^*$) decreases (increases) with $p_c$. As more money be paid in the carbon market, manufacturer (supplier) will reduce (increase) ordering quantity (wholesale price) for profit. From Figure 7a,d, we observe that $q^*$ under model *tt* is the lowest, which presents that manufacturer chooses greatly reduce $q^*$ to reduce the risk of loss when two firms are face financial constraints.

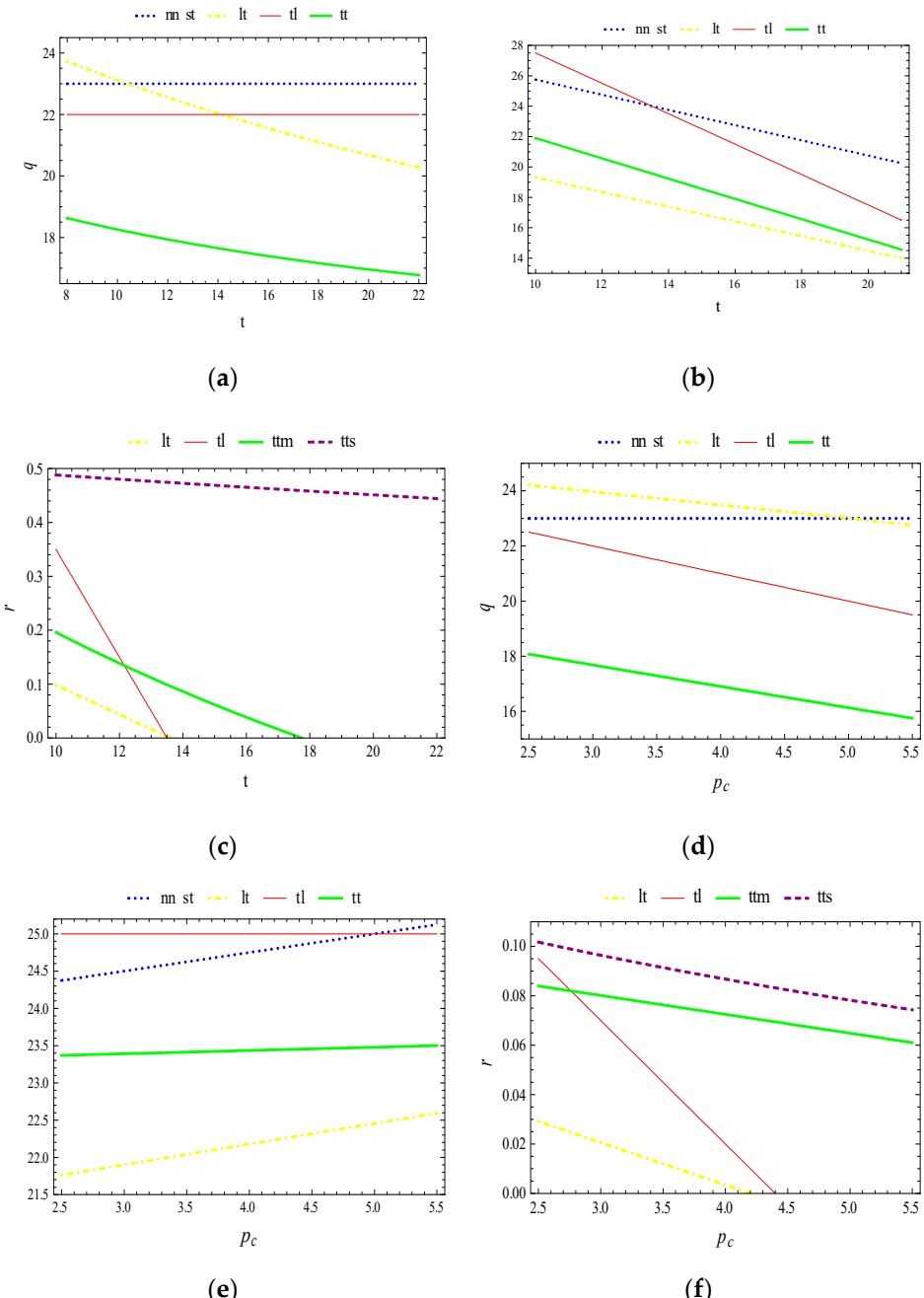

**Figure 7.** The impacts of $t$ and $p_c$ on optimal operational strategies. (**a**) Effect of $t$ on $q^*$ (**b**) Effect of $t$ on $\omega^*$ (**c**) Effect of $t$ on $r^*$ (**d**) Effect of $p_c$ on $q^*$ (**e**) Effect of $p_c$ on $\omega^*$ (**f**) Effect of $p_c$ on $r^*$.

It can be seen from Figure 7c,f that optimal loan interest rate $r^*$ decreases with $t$ ($p_c$), and $r^*$ under model *tt* decreases more moderately than that under model *lt* and *tl*. The 3PL's interest rate to the manufacturer (supplier) under model *lt* (*tl*) is less than that under model *tt*. With the increasing of $t$ ($p_c$), 3PL needs to reduce $r^*$ to improve the financing motivation of members, and in order to decrease a great loss risk, the 3PL will set a relatively high interest rate under model *tt*. Since manufacturer is forced to bear excessive costs when $t$ ($p_c$) too high, which may lead to capital-constrained manufacturer or supplier abandons financing, namely, $r^*$ is 0 if $t$ ($p_c$) is larger than a constant value. The supplier as a leader can obtain more profitable opportunities, this is why 3PL always set a higher $r^*$ to the supplier compare with the manufacturer.

As shown in Figure 8, we investigate the impacts of carbon emission from manufacturer and supplier $e_0$, $e_1$ on supply chain performance. It can be seen from Figure 8a,c that $q^*$ under model *lt* is larger than that under model *nn* and *st* if $e_0$ or $e_1$ is enough low. And $q^*$ under model *tl* is less than that under model *nn* and *st*, which means that capital-constrained supplier is harmful to $q^*$, and 3PL financing cannot make up for this loss. From Figure 8b,d, we know that the 3PL's interest rate to the manufacturer (supplier) decreases with $e_0$. The increasing of $e_0$ leads to the profits of supplier and manufacturer are reduced, thus, the 3PL needs to decrease $r^*$ to protect financing business. In addition, as $e_1$ increases, the 3PL will set a higher (lower) interest rate to the manufacturer (supplier) to reduce loss risk (improve financing business).

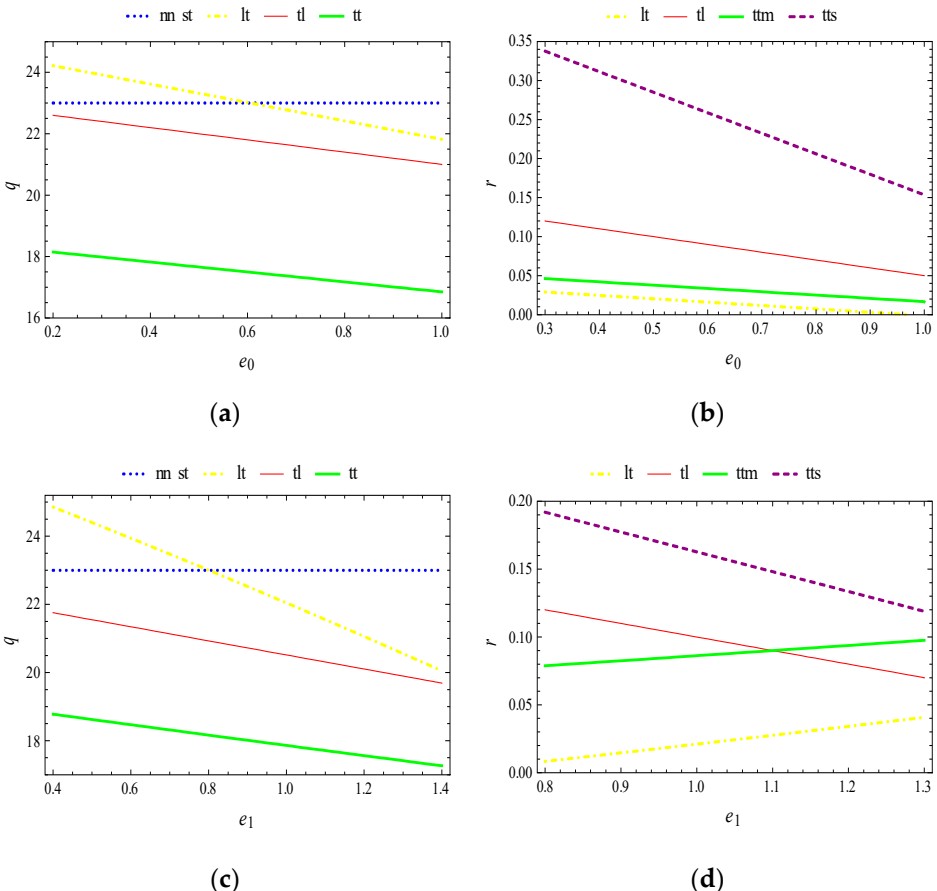

**Figure 8.** The impacts of $e_0$ and $e_1$ on optimal operational strategies. (**a**) Effect of $e_0$ on $q^*$ (**b**) Effect of $e_0$ on $r^*$ (**c**) Effect of $e_1$ on $q^*$ (**d**) Effect of $e_1$ on $r^*$.

As shown in Figure 9, we consider the impacts of the initial capital of manufacturer and supplier $b$, $s$ on the operational strategies of supply chain members. From Figure 9a, we observe that $q^*$ increase with $b$, and $q^*$ under model *lt* is less than that under model *nn* and *st* when $b$ is less than 296. The increasing of $b$ can reduce manufacturer's loss risk, which allows manufacturer to accept a higher $\omega^*$. Figure 9b illustrates that 3PL will determine a lower (higher) interest rate to the manufacturer (supplier) as $b$ increases, which can obtain new interests from financing business. Similarly, we can observe from Figure 9c that $q^*$ under model *tl* is less than that under model *nn*, *st* and *lt* when $s$ is less than 60, and $q^*$ increases with $s$. Figure 9d presents that 3PL will determine a higher (lower) loan interest rate to the supplier (manufacturer) as $s$ increases, and 3PL's interest rate to the manufacturer (supplier) is 0 if the $b$ ($s$) is enough high.

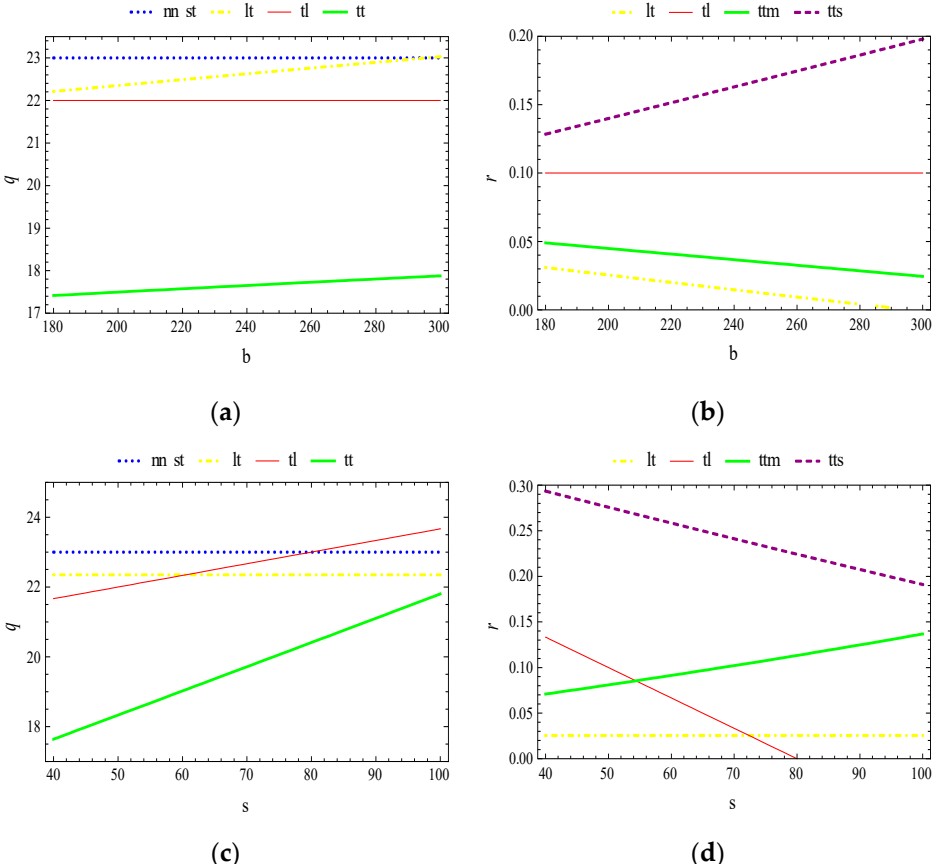

**Figure 9.** The impacts of $b$ and $s$ on optimal operational strategies. (**a**) Effect of $b$ on $q^*$ (**b**) Effect of $b$ on $r^*$ (**c**) Effect of $s$ on $q^*$ (**d**) Effect of $s$ on $r^*$.

### 7.2. Impact of Relevant Parameters on Optimal Profits

As shown in Figure 10, we investigate the impacts of 3PL's transportation fee $t$ and carbon trading price $p_c$ on the profits of supply chain members. From Figure 10a,b, we find that $\Pi^{m^*}$ and $\Pi^{s^*}$ decrease with $t$. It can be seen from Figure 10c that $\Pi^{l^*}$ increases with $t$ when $t$ is less than a constant value, it then decreases with $t$. This is because $\Pi^{l^*}$ can be increased due to the increased profit of financing business when $t$ is enough low. However, a higher $t$ is harmful to $q^*$ and may lead to a capital-constrained manufacturer goes bankruptcy, which means that adopts a too high $t$ is not good for the development of supply chain, thus, the 3PL should reasonably adjust $t$ to stimulate $q^*$. From Figure 10d,e, we observe that $\Pi^{m^*}$ and $\Pi^{s^*}$ increase with $p_c$, but $\Pi^{l^*}$ decreases with $p_c$. The manufacturer and supplier can make more profits through trading carbon emission cap, but $\Pi^{l^*}$ is damaged by the reduction of $q^*$.

We compare $\Pi^{m*}$, $\Pi^{s*}$ and $\Pi^{l*}$ under variable scenarios, respectively. From Figure 10a,b,d,e, we observe that $\Pi^{m*}$ ($\Pi^{s*}$) under model *tt* is larger (lower) than that under other models, which means that financial constraint benefits (harms) the manufacturer (supplier) when 3PL provides financing service. We also find that $\Pi^{m*}$ under model *lt* is larger than that under model *nn*, *st* and *tl* if $t$ is enough low, and $\Pi^{s*}$ under model *tl* is always less than that under model *nn* and *st* regardless of the value of $t$ or $p_c$. Under 3PL financing mode, manufacturer with financial constraint has a larger profit than one without financial constrain when $t$ is less than a certain constant, and supplier with financial constrain has a lower profit than one without financial constrain. From Figure 10c,f, we observe that $\Pi^{l*}$ under model *tt* is larger than that under other models and $\Pi^{l*}$ under model *nn* and *st* are lower than that under model *lt* and *tl* if $t$ is enough low, which means that 3PL should incorporate financing service into its business and offer a lower $t$ to capital-constrained members. Since 3PL under model *lt* always

has more profits than that under model *tl*, we conclude that a capital-constrained manufacturer is more beneficial to 3PL than capital-constrained supplier.

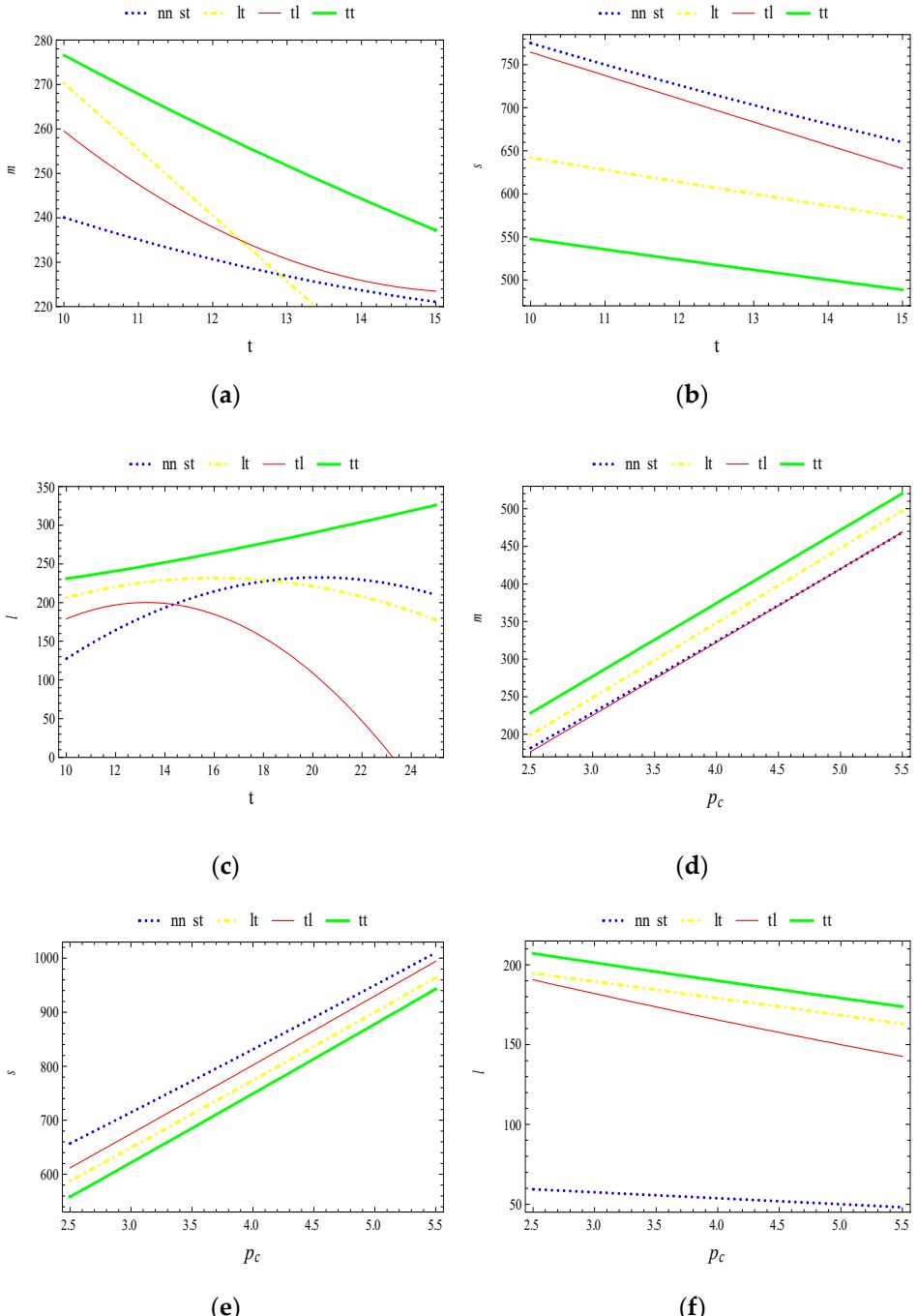

**Figure 10.** The impacts of $t$ and $p_c$ on members' profits. (**a**) Effect of $t$ on $\Pi^{m*}$ (**b**) Effect of $t$ on $\Pi^{s*}$ (**c**) Effect of $t$ on $\Pi^{l*}$ (**d**) Effect of $p_c$ on $\Pi^{m*}$ (**e**) Effect of $p_c$ on $\Pi^{s*}$ (**f**) Effect of $p_c$ on $\Pi^{l*}$.

As shown in Figure 11, we consider the impacts of carbon emission from manufacturer and supplier $e_0$, $e_1$ on the profits of supply chain members. From Figure 11a,c, we find that the manufacturer's (supplier's) profit decreases with $e_0$ ($e_1$). Thus, it is necessary to reduce carbon emissions for an emission-dependent supply chain. Figure 11b,d present that $e_0$ and $e_1$ also have negative effects on $\Pi^{l*}$, thus, the 3PL should encourage partners to reduce operational carbon emissions. The supplier under model *tl* has more profits than that under model *lt*, which means that if 3PL offers financing,

the supplier with financial constraint benefits supplier compare with the manufacturer with financial constraint. Interestingly, we corroborate that, if $e_0$ is less (more) than a constant value, the manufacturer with financial constraint benefits (harms) 3PL firm compared with the supplier with financial constraint. And if $e_1$ is less (more) than a constant value, the supplier with financial constraint benefits (harms) 3PL firm compared with the manufacturer with financial constraint.

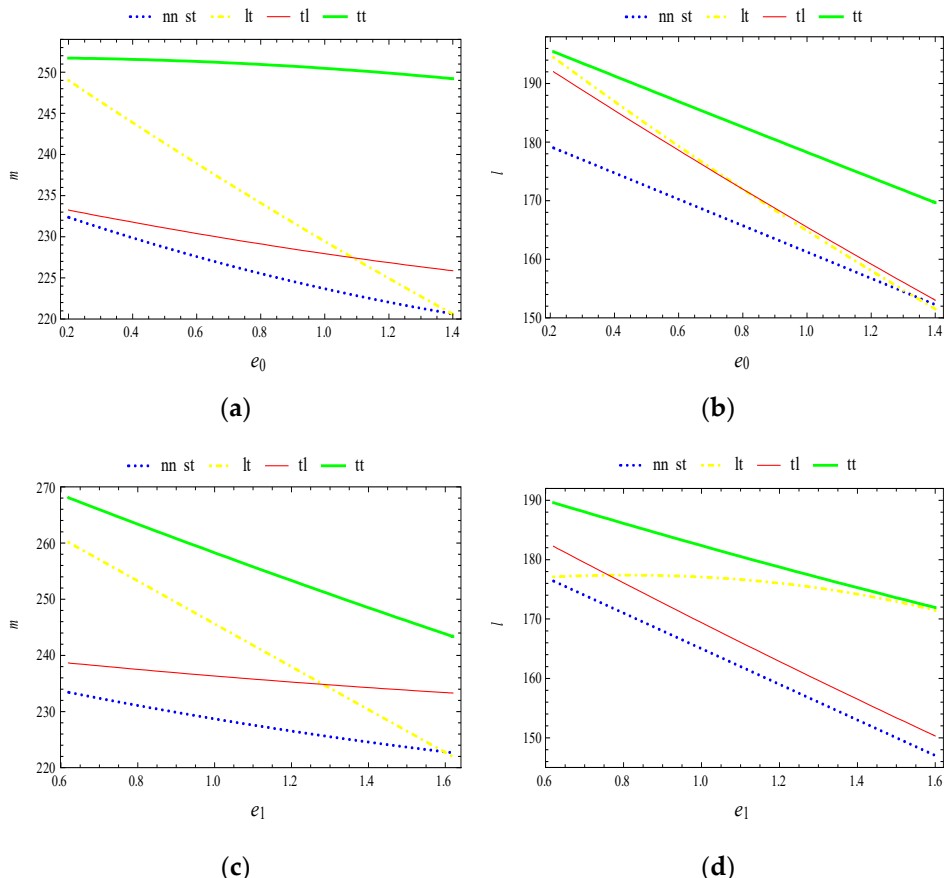

**Figure 11.** The impacts of $e_0$ and $e_1$ on members' profits. (**a**) Effect of $e_0$ on $\Pi^{m*}$ (**b**) Effect of $e_0$ on $\Pi^{l*}$ (**c**) Effect of $e_1$ on $\Pi^{s*}$ (**d**) Effect of $e_1$ on $\Pi^{l*}$.

As shown in Figure 12 and Table 2, we consider the impacts of the initial capital of manufacturer and supplier $b, s$ on the profits of supply chain members. A higher initial capital can decrease the loss risk and obtain more profitable opportunities, thus, $\Pi^{m*}$ and $\Pi^{s*}$ increase as $b$ and $s$ increase. And the manufacturer's (supplier's) profit under model $lt$ ($tl$) is larger than that under model $tl$ ($lt$), $nn$ and $st$ when $b$ ($s$) is enough high. According to Table 2, it is obvious to see that $\Pi^{l*}$ under model $lt$ and $tt$ increases with $b$ firstly and then decreases with $b$. This is because $\Pi^{l*}$ is composed of transportation profit and financing revenue, so the relationship between $\Pi^{l*}$ and $b$ can be depicted as an inverse U-shaped curve. But $\Pi^{l*}$ always increases with the increasing of $s$. It implies that 3PL does not always like (more prefer) to cooperate with a rich manufacturer (supplier). Cooperating with a medium rich manufacturer may bring more profit.

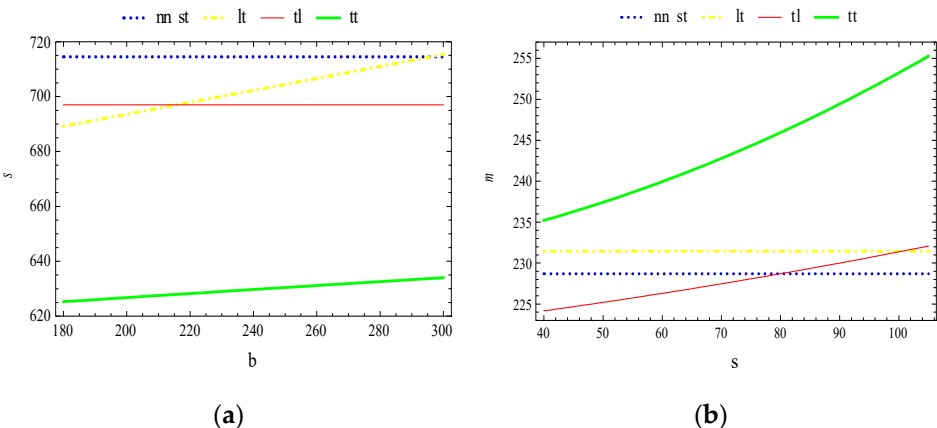

**Figure 12.** The impacts of *b* and *s* on members' profits. (**a**) Effect of *b* on $\Pi^{s*}$ (**b**) Effect of *s* on $\Pi^{m*}$.

**Table 2.** The Optimal Profit of 3PL Firm.

| | 3PL Firm's Profit | | | | | 3PL Firm's Profit | | | |
|---|---|---|---|---|---|---|---|---|---|
| *b* | *nn/st* | *lt* | *tl* | *tt* | *s* | *nn/st* | *lt* | *tl* | *tt* |
| 100 | 172.5 | 146.29 | 182 | 172.012 | 10 | 172.5 | 183.261 | 174.667 | 184.729 |
| 120 | 172.5 | 146.329 | 182 | 172.03 | 20 | 172.5 | 183.261 | 176.5 | 190.693 |
| 140 | 172.5 | 146.359 | 182 | 172.035 | 30 | 172.5 | 183.261 | 178.333 | 197.187 |
| 160 | 172.5 | 146.378 | 182 | 172.029 | 40 | 172.5 | 183.261 | 180.167 | 204.21 |
| 180 | 172.5 | 146.387 | 182 | 172.012 | 50 | 172.5 | 183.261 | 182 | 211.756 |
| 200 | 172.5 | 146.386 | 182 | 171.983 | 60 | 172.5 | 183.261 | 183.833 | 219.823 |
| 220 | 172.5 | 146.376 | 182 | 171.943 | 70 | 172.5 | 183.261 | 185.667 | 228.405 |
| 240 | 172.5 | 146.356 | 182 | 171.893 | 80 | 172.5 | 183.261 | 187.5 | 237.499 |
| 260 | 172.5 | 146.327 | 182 | 171.832 | 90 | 172.5 | 183.261 | 189.333 | 247.099 |
| 280 | 172.5 | 146.289 | 182 | 171.76 | 100 | 172.5 | 183.261 | 191.167 | 257.201 |

*7.3. Selection of Financing Mode for the Manufacturer and Supplier*

From Figure 13a,c, we observe that $\Pi^{m*}$ under model *lt* is highest when *t* is less than a constant value and $\Pi^{m*}$ under model *st* is always higher than that under model *bt*. Thus, the manufacturer will choose supplier financing mode when *t* is higher than a certain threshold, and this certain threshold increases with $e_0$. It can be seen from Figure 13b,d that $\Pi^{s*}$ under model *nn* is highest, which means that the 3PL or bank financing mode can't coordinate this supply chain in this situation. The $\Pi^{s*}$ under model *tl* is higher than that under model *bt* when *t* is enough low, and this certain threshold decreases with $e_1$. This phenomenon demonstrates that *t*, $e_0$ and $e_1$ have important influences on the selection of financing mode for the manufacturer and supplier.

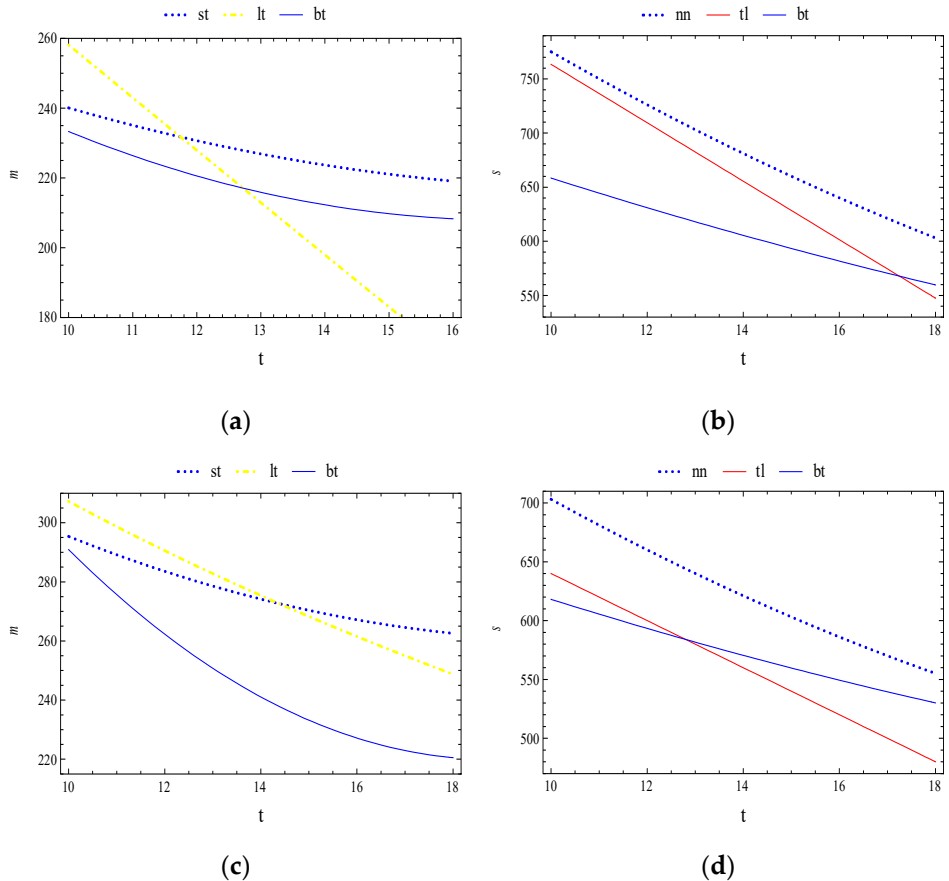

**Figure 13.** The impacts of $t$, $e_0$ and $e_1$ on members' profits. (**a**) Effect of $t$ on $\Pi^{m*}$ ($e_0 = 0.5$) (**b**) Effect of $t$ on $\Pi^{s*}$ ($e_1 = 1$) (**c**) Effect of $t$ on $\Pi^{m*}$ ($e_0 = 1.5$) (**d**) Effect of $t$ on $\Pi^{s*}$ ($e_1 = 2$).

## 7.4. Impact of Manufacturer's Loss Aversion

From Figure 14a,b, we can see that the profit of each supply chain member increases with the manufacturer's carbon cap $M$. If the government sets a higher $M$, manufacturer can obtain more profits by trading emission caps, which also benefits other partners. And the manufacturer's loss aversion $\lambda$ is harmful (beneficial) to manufacturer and 3PL firm (supplier). The $\lambda$ has a negative effect on $q^*$, namely, a capital-constrained manufacturer prefers to order less to make a little profit with a lower loss risk rather than facing a high loss risk. In addition, the bankruptcy threshold of capital-constrained manufacturer decreases (increases) with $M$ ($\lambda$), and with an increase in $\lambda$, the required $M$ increases.

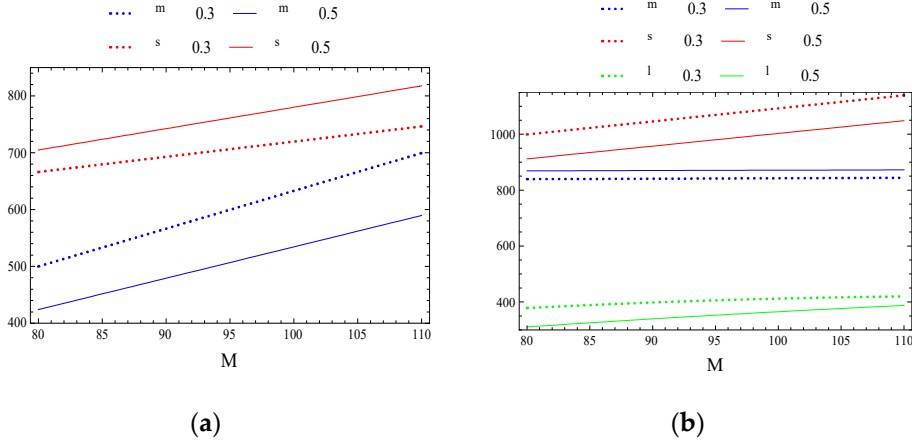

**Figure 14.** The impacts of $M$ and $\lambda$ on members' profits. (**a**) Effect of $M$ on $\Pi^*_{sa}$ (**b**) Effect of $M$ on $\Pi^*_{la}$.

## 8. Conclusions

In order to curb carbon emissions, many countries have implemented cap-and-trade policy. In this context, we introduce the member's financial situation into an emission-dependent supply chain. With the increasing intensity of competition, 3PL chooses to expand their financing business. There are very few researches focused on the other roles of 3PL firm. Based on this situation, we assume that 3PL offers variable transportation fee strategy and financing service for capital-constrained members under the stochastic market demand. This paper considers a three-echelon supply chain and formulates five different financing models on the basis of manufacturer and supplier with or without capital constraints ($nn, st, lt, tl, tt$). The impacts of cap-and-trade system and variable transportation fee strategy on the supply chain performance are analyzed. We also discuss the optimal selection of financing mode for the manufacturer and supplier.

The major findings can be summarized as follows.

(1)  The carbon emissions of members are harmful to ordering quantity and loan interest rate, so manufacturer and supplier should adopt advanced technology to reduce operational emissions. In addition, the 3PL and government should encourage this behavior and take into account the firm's financing situation when implementing the strategies.

(2)  The 3PL's variable transportation fee strategy is necessary operation in the supply chain practice. The increasing of transportation fee will result in a reduction in ordering quantity and loan interest rate. The ordering quantity and profit of manufacturer under the 3PL financing service mode are larger than those under the two firms are well-funded model when transportation fee is less than a certain constant. But the 3PL's profit increases with the transportation fee firstly, then decreases with it. Therefore, 3PL should reasonably adjust the transportation fee strategy and interest rate decisions while protecting its own profits.

(3)  The 3PL's loan interest rate to the manufacturer and supplier are related to many factors. The increasing of retailer price and salvage value of unsold products can reduce the 3PL's loan interest rate. Higher capital constrain will stimulate 3PL to adopt a higher interest rate to the manufacturer (supplier). And if both two firms have financial constraints, the 3PL will set a relatively higher interest rate to control the loss risk. However, the 3PL's loan interest rate to the manufacturer (supplier) will become 0 when the transportation fee or members' initial carbon emissions (capitals) are larger than a threshold.

(4)  The impact of each member's financial situation on the supply chain performance is variable. Under the 3PL financing service mode, the capital constraint is beneficial (harmful) to manufacturer (supplier). If 3PL sets a suitable transportation fee and interest rate, the manufacturer with financial constraint has a larger profit than one without financial constraint. On the contrary, the 3PL financing mode cannot fill the supplier's funding gap. For the 3PL firm, it can obtain additional profit due to the development of financing business, and more willing to offer financing support for the manufacturer. If the manufacturer's (supplier's) carbon emission is less (larger) than a constant value, the manufacturer (supplier) with financial constraint benefits (harms) the 3PL compared with the supplier (manufacturer) with financial constraint. In order to obtain more profits, 3PL prefers to cooperate with a medium rich (wealthy) manufacturer (supplier).

(5)  From the comparisons of different financing modes, we find that a capital-constrained manufacturer (supplier) will choose 3PL (bank) financing mode to obtain the highest profit when transportation fee is lower (larger) than a certain threshold, and this certain threshold increases (decreases) with the manufacturer's (supplier's) carbon emission. However, the 3PL or bank financing mode can't coordinate the supply chain when supplier has financial constraint. The manufacturer's loss aversion (carbon cap) can increase (decrease) its bankruptcy threshold, and more loss-averse manufacturer requires a larger carbon cap.

Our research explores the 3PL financing service mode in an emission-dependent supply chain under the cap-and-trade system, which opens a new direction for future studies. First, this paper only

considers the carbon emissions of manufacturer and supplier, thus it is a future research direction to consider the carbon emission behavior of 3PL firm. Second, the information is assumed as shared knowledge for firms in this paper, thus it is interesting to investigate the 3PL financing modes in an information asymmetry scenario.

**Author Contributions:** C.L. conceived the model and designed the experiments; C.L. and K.Z. wrote the paper; J.Y. revised the whole details.

**Funding:** This research is supported by the National Natural Science Foundation of China (No.71472183).

**Acknowledgments:** The authors are grateful to the editors and anonymous reviewers for their kindly review and helpful comments.

**Conflicts of Interest:** The authors declare no conflict of interest.

## Appendix A. Proposition 1–7

**Proposition 1.** *(1) For a given supplier's wholesale price $\omega_{nn}$, the well-funded manufacturer's optimal ordering quantity satisfies $q_{nn}^* = \overline{F}^{-1}(\frac{\omega_{nn}+t_{nn}+p_c e_0 - \varepsilon}{p-\varepsilon})$.*

*(2) When the supplier and manufacturer are without capital constraints, the supplier's expected utility is concave in $\omega_{nn}$ and the optimal wholesale price satisfies $\omega_{nn}^* = \frac{H(q_{nn}^*(\omega_{nn}^*))(t_{nn}+p_c e_0 - \varepsilon)+c+p_c e_1}{1-H(q_{nn}^*(\omega_{nn}^*))}$.*

**Proposition 2.** *(1) Given the supplier's wholesale price and loan interest rate $\omega_{st}$ and $r_{st}$, the capital-constrained manufacturer's optimal ordering quantity is uniquely given by $q_{st}^* = \overline{F}^{-1}(\frac{(\omega_{st}+t_{st})(1+r_{st})+p_c e_0 - \varepsilon}{p-\varepsilon})$.*

*(2) When the supplier provides financing service to manufacturer, the supplier's optimal loan interest rate $r_{st}^* = 0$ and wholesale price satisfies $\omega_{st}^* = \frac{H(q_{st}^*(\omega_{st}^*))t_{st}+c+p_c e_1+(p_c e_0 - \varepsilon)H(q_{st}^*(\omega_{st}^*))}{1-H(q_{st}^*(\omega_{st}^*))}$.*

**Proposition 3.** *(1) Given the supplier's wholesale price $\omega_{lt}$ and 3PL's loan interest rate $r_{lt}$, the manufacturer's optimal ordering quantity satisfies $q_{lt}^* = \overline{F}^{-1}(\frac{(\omega_{lt}+t_{lt})(1+r_{lt})+p_c e_0 - \varepsilon}{p-\varepsilon})$.*

*(2) For a given supplier's wholesale price $\omega_{lt}$, the 3PL firm's optimal loan rate*

*satisfies* $r_{lt}^* = \begin{cases} \widetilde{r_{lt}}, 0 \le b < \frac{((\omega_{lt}+t_{lt})q_{lt}^*(r_{lt})-b)(p-\varepsilon)f(q_{lt}^*(r_{lt}))-(\omega_{lt}+t_{lt})(t_{lt}-n)}{(p-\varepsilon)f(q_{lt}^*(r_{lt}))} \\ 0, b \ge \frac{((\omega_{lt}+t_{lt})q_{lt}^*(r_{lt})-b)(p-\varepsilon)f(q_{lt}^*(r_{lt}))-(\omega_{lt}+t_{lt})(t_{lt}-n)}{(p-\varepsilon)f(q_{lt}^*(r_{lt}))} \end{cases}$, *where* $\widetilde{r_{lt}} = \frac{((\omega_{lt}+t_{lt})q_{lt}^*(r_{lt})-b)(p-\varepsilon)f(q_{lt}^*(r_{lt}))-(\omega_{lt}+t_{lt})(t_{lt}-n)}{(\omega_{lt}+t_{lt})^2}$.

*(3) When the 3PL provides financing service to manufacturer, the supplier's optimal wholesale price is uniquely given by $\omega_{lt}^* = \frac{(1+r_{lt}^*(\omega_{lt}^*))(H(q_{lt}^*(\omega_{lt}^*))t_{lt}+c+p_c e_1)+(p_c e_0 - \varepsilon)H(q_{lt}^*(\omega_{lt}^*))}{(1-H(q_{lt}^*(\omega_{lt}^*)))(1+r_{lt}^*(\omega_{lt}^*))}$.*

**Proposition 4.** *(1) For a given supplier's wholesale price $\omega_{tl}$, the manufacturer's optimal ordering quantity satisfies $q_{tl}^* = \overline{F}^{-1}(\frac{\omega_{tl}+t_{tl}+p_c e_0 - \varepsilon}{p-\varepsilon})$.*

*(2) Given the 3PL's loan interest rate $r_{tl}$, the supplier's optimal wholesale price satisfies $\omega_{tl}^* = \frac{H(q_{tl}^*(\omega_{tl}^*))(t_{tl}+p_c e_0 - \varepsilon)+p_c e_1+c(1+r_{tl})}{1-H(q_{tl}^*(\omega_{tl}^*))}$.*

*(3) When the 3PL firm provides financing service to supplier, the 3PL firm's optimal loan interest rate satisfies $r_{tl}^* = \frac{c(n-t_{tl})-f(q_{tl}^*(r_{tl}^*))(p-\varepsilon)(s-cq_{tl}^*(r_{tl}^*))}{c^2}$.*

**Proposition 5.** *(1) Given the supplier's wholesale price $\omega_{tt}$ and 3PL firm's loan interest rate $r_m$, the manufacturer's optimal ordering quantity satisfies $q_{tt}^* = \overline{F}^{-1}(\frac{(\omega_{tt}+t_{tt})(1+r_m)+p_c e_0 - \varepsilon}{p-\varepsilon})$.*

*(2) Given the 3PL firm's loan interest rate $r_m$ and $r_s$, the supplier's optimal wholesale price satisfies $\omega_{tt}^* = \frac{(1+r_m)(H(q_{tt}^*(\omega_{tt}^*))t_{tt}+c(1+r_s)+p_c e_1)+(p_c e_0 - \varepsilon)H(q_{tt}^*(\omega_{tt}^*))}{(1-H(q_{tt}^*(\omega_{tt}^*)))(1+r_m)}$.*

*(3) When the 3PL firm provides financing service to supplier and manufacturer, the 3PL's optimal loan interest rate to the manufacturer satisfies* $r_m^* = \begin{cases} \widetilde{r_m},\ 0 \le b < \frac{((\omega_{tt}^*(r_m)+t_{tt})q_{tt}^*(r_m)-b)(p-\varepsilon)f(q_{tt}^*(r_m))-(\omega_{tt}^*(r_m)+t_{tt})(t_{tt}-n)}{(p-\varepsilon)(\omega_{tt}^*(r_m)+t_{tt})^2} \\ 0,\ b \ge \frac{((\omega_{tt}^*(r_m)+t_{tt})q_{tt}^*(r_m)-b)(p-\varepsilon)f(q_{tt}^*(r_m))-(\omega_{tt}^*(r_m)+t_{tt})(t_{tt}-n)}{(p-\varepsilon)(\omega_{tt}^*(r_m)+t_{tt})^2} \end{cases}$,

*where* $\widetilde{r_m} = \frac{((\omega_{tt}^*(r_m)+t_{tt})q_{tt}^*(r_m)-b)(p-\varepsilon)f(q_{tt}^*(r_m))-(\omega_{tt}^*(r_m)+t_{tt})(t_{tt}-n)}{(\omega_{tt}^*(r_m)+t_{tt})^2}$. *And the 3PL's optimal loan interest rate to the supplier satisfies* $r_s^* = \frac{f(q_{tt}^*(r_s))(p-\varepsilon)(cq_{tt}^*(r_s)-s)}{c^2(1+r_m)}$.

**Proposition 6.** *Solving the model through backward induction, we can obtain the optimal ordering quantity as* $q_{sa}^* = \overline{F}^{-1}\left(\frac{(\omega_{sa}+t_{sa}+p_ce_0-\varepsilon)(\lambda-(\lambda-1)\overline{F}(x_0))}{(p-\varepsilon)}\right)$, *the optimal interest rate as* $r_{sa}^* = 0$, *and the optimal wholesale price as* $\omega_{sa}^* = \frac{F(x_0)(t_{sa}-p_ce_0-\varepsilon)+c+p_ce_1-\overline{F}(x_0)q_{sa}^*(\omega_{sa})(d\omega_{sa}/dq_{sa}^*(\omega_{sa}))}{\overline{F}(x_0)}$.

**Proposition 7.** *Similar to the analysis in Section 5.2.1, we can obtain the optimal ordering quantity as* $q_{la}^* = \overline{F}^{-1}\left(\frac{((\omega_{la}+t_{la})(1+r_{la})+p_ce_0-\varepsilon)(\lambda-(\lambda-1)\overline{F}(x_0))}{(p-\varepsilon)}\right)$, *the optimal wholesale price as* $\omega_{la}^* = c + p_ce_1 - \frac{q_{la}^*(\omega_{la})}{(dq_{la}^*(\omega_{la})/d\omega_{la})}$, *and the optimal interest rate as* $r_{la}^* = \frac{F(x_0)(\omega_{la}+t_{la}-p_ce_0-\varepsilon)+n-t_{la}-\overline{F}(x_1)((\omega_{la}+t_{la})q_{la}^*(r_{la})-b)(dr_{la}/dq_{la}^*(r_{la}))}{(\omega_{la}+t_{la})\overline{F}(x_1)}$.

## Appendix B. Proof of Proposition

**Proof of Proposition 1.** (1) According to Equation (1), taking the first-order and second-order derivatives of $\Pi_{nn}^m(q_{nn})$ with respect to $q_{nn}$, we can get $\frac{d\Pi_{nn}^m(q_{nn})}{dq_{nn}} = (p-\varepsilon)\overline{F}(q_{nn}) + \varepsilon - p_ce_0 - \omega_{nn} - t_{nn}$, $\frac{d^2\Pi_{nn}^m(q_{nn})}{dq_{nn}^2} = -(p-\varepsilon)f(q_{nn}) < 0$. Therefore, $\Pi_{nn}^m(q_{nn})$ is concave in $q_{nn}$. According to the first-order condition of $\frac{d\Pi_{nn}^m(q_{nn})}{dq_{nn}} = 0$, we get $q_{nn}^* = \overline{F}^{-1}\left(\frac{\omega_{nn}+t_{nn}+p_ce_0-\varepsilon}{p-\varepsilon}\right)$.

(2) According to Equation (2), taking the first-order and second-order derivatives of $\Pi_{nn}^s(\omega_{nn})$ with respect to $\omega_{nn}$, we can get $\frac{d\Pi_{nn}^s(\omega_{nn})}{d\omega_{nn}} = q_{nn}^*(\omega_{nn}) - \frac{\omega_{nn}-c-p_ce_1}{f(q_{nn}^*(\omega_{nn}))(p-\varepsilon)}$, $\frac{d^2\Pi_{nn}^s(\omega_{nn})}{d\omega_{nn}^2} = \frac{dq_{nn}^*(\omega_{nn})}{d\omega_{nn}}\left(\frac{2(p-\varepsilon)f^2(q_{nn}^*(\omega_{nn}))+(\omega_{nn}-c-p_ce_1)f'(q_{nn}^*(\omega_{nn}))}{f^2(q_{nn}^*(\omega_{nn}))(p-\varepsilon)}\right)$,

(i) If $f'(q_{nn}^*(\omega_{nn})) \ge 0$, then we have $\frac{d^2\Pi_{nn}^s(\omega_{nn})}{d\omega_{nn}^2} < 0$,

(ii) If $f'(q_{nn}^*(\omega_{nn})) < 0$, since $\frac{\omega_{nn}-c-p_ce_1}{2(p-\varepsilon)} < \frac{\omega_{nn}+t_{nn}+p_ce_0-\varepsilon}{p-\varepsilon} = \overline{F}(q_{nn}^*(\omega_{nn}))$, thus,

$\frac{d^2\Pi_{nn}^s(\omega_{nn})}{d\omega_{nn}^2} = \frac{dq_{nn}^*(\omega_{nn})}{d\omega_{nn}}\left(\frac{2(p-\varepsilon)f^2(q_{nn}^*(\omega_{nn}))+(\omega_{nn}-c-p_ce_1)f'(q_{nn}^*(\omega_{nn}))}{f^2(q_{nn}^*(\omega_{nn}))(p-\varepsilon)^2}\right)$

$< 2\frac{dq_{nn}^*(\omega_{nn})}{d\omega_{nn}}\left(\frac{f^2(q_{nn}^*(\omega_{nn}))+\overline{F}(q_{nn}^*(\omega_{nn}))f'(q_{nn}^*(\omega_{nn}))}{f^2(q_{nn}^*(\omega_{nn}))}\right)$

$= 2\frac{dq_{nn}^*(\omega_{nn})}{d\omega_{nn}}\left(\frac{h'(q_{nn}^*(\omega_{nn}))\overline{F}(q_{nn}^*(\omega_{nn}))}{f^2(q_{nn}^*(\omega_{nn}))}\right) < 0$

It is obvious to conclude that $\Pi_{nn}^s(\omega_{nn})$ is concave in $\omega_{nn}$. According to the first-order condition of $\frac{d\Pi_{nn}^s(\omega_{nn})}{d\omega_{nn}} = 0$, we get $\omega_{nn}^* = \frac{H(q_{nn}^*(\omega_{nn}))(t_{nn}+p_ce_0-\varepsilon)+c+p_ce_1}{1-H(q_{nn}^*(\omega_{nn}))}$. □

**Proof of Proposition 2.** (1) According to Equation (3), similar to the analysis in the Proof of Proposition 1, we know that the second-order condition $\frac{d^2\Pi_{st}^m(q_{st})}{dq_{st}^2} < 0$ holds, so $\Pi_{st}^m(q_{st})$ is concave in $q_{st}$. Let $\frac{d\Pi_{st}^m(q_{st})}{dq_{st}} = 0$, we can get $q_{st}^* = \overline{F}^{-1}\left(\frac{(\omega_{st}+t_{st})(1+r_{st})+p_ce_0-\varepsilon}{p-\varepsilon}\right)$.

(2) According to Equation (4), similar to the analysis in the Proof of Proposition 1, we can obtain the optimal loan interest rate $r_{st}^* = 0$ and $\Pi_{st}^s(\omega_{st})$ is concave in $\omega_{st}$. From the first-order condition of $\frac{d\Pi_{st}^s(\omega_{st})}{d\omega_{st}} = 0$, we can obtain $\omega_{st}^* = \frac{H(q_{st}^*(\omega_{st}^*))(t_{st}+p_ce_0-\varepsilon)+c+p_ce_1}{1-H(q_{st}^*(\omega_{st}^*))}$. □

**Proof of Proposition 3.** (1) According to Equation (5), similar to the analysis in the Proof of Proposition 1, we can get $\frac{d\Pi_{lt}^m(q_{lt})}{dq_{lt}} = (p - \varepsilon)\overline{F}(q_{lt}) + \varepsilon - p_c e_0 - (\omega_{lt} + t_{lt})(1 + r_{lt})$, $\frac{d^2\Pi_{lt}^m(q_{lt})}{dq_{lt}^2} = -(p - \varepsilon)f(q_{lt}) < 0$.

As a result, let $\frac{d\Pi_{lt}^m(q_{lt})}{dq_{lt}} = 0$, we can get $q_{lt}^* = \overline{F}^{-1}\left(\frac{(\omega_{lt}+t_{lt})(1+r_{lt})+p_c e_0-\varepsilon}{p-\varepsilon}\right)$.

(2) According to Equation (6), taking the first-order and second-order derivatives of $\Pi_{lt}^s(r_{lt})$ with respect to $r_{lt}$, we can get $\frac{d\Pi_{lt}^s(r_{lt})}{dr_{lt}} = (\omega_{lt} + t_{lt})\left(q_{lt}^*(r_{lt}) - \frac{(\omega_{lt}+t_{lt})r_{lt}+t_{lt}-n}{f(q_{lt}^*(r_{lt}))(p-\varepsilon)}\right) - b$, $\frac{d^2\Pi_{lt}^s(r_{lt})}{dr_{lt}^2} = (\omega_{lt} + t_{lt})\frac{dq_{lt}^*(r_{lt})}{dr_{lt}}\left(\frac{2(p-\varepsilon)f^2(q_{lt}^*(r_{lt}))+((\omega_{lt}+t_{lt})r_{lt}+t_{lt}-n)f'(q_{lt}^*(r_{lt}))}{f^2(q_{lt}^*(r_{lt}))(p-\varepsilon)}\right)$,

(i) If $f'(q_{lt}^*(r_{lt})) \geq 0$, then we have $\frac{d^2\Pi_{lt}^s(r_{lt})}{dr_{lt}^2} < 0$,

(ii) If $f'(q_{lt}^*(r_{lt})) < 0$, since $\frac{(\omega_{lt}+t_{lt})r_{lt}+t_{lt}-n}{2(p-\varepsilon)} < \frac{(\omega_{lt}+t_{lt})(1+r_{lt})+p_c e_0-\varepsilon}{p-\varepsilon} = \overline{F}(q_{lt}^*(r_{lt}))$, thus,

$$\frac{d^2\Pi_{lt}^s(r_{lt})}{dr_{lt}^2} = (\omega_{lt} + t_{lt})\frac{dq_{lt}^*(r_{lt})}{dr_{lt}}\left(\frac{2(p-\varepsilon)f^2(q_{lt}^*(r_{lt}))+((\omega_{lt}+t_{lt})r_{lt}+t_{lt}-n)f'(q_{lt}^*(r_{lt}))}{f^2(q_{lt}^*(r_{lt}))(p-\varepsilon)}\right)$$
$$< 2(\omega_{lt} + t_{lt})\frac{dq_{lt}^*(r_{lt})}{dr_{lt}}\left(\frac{f^2(q_{lt}^*(r_{lt}))+\overline{F}(q_{lt}^*(r_{lt}))f'(q_{lt}^*(r_{lt}))}{f^2(q_{lt}^*(r_{lt}))}\right)$$
$$= 2(\omega_{lt} + t_{lt})\frac{dq_n^*(r_{lt})}{dr_{lt}}\left(\frac{h'(q_{lt}^*(r_{lt}))\overline{F}(q_{lt}^*(r_{lt}))}{f^2(q_{lt}^*(r_{lt}))}\right) < 0$$

It is obvious to conclude that $\Pi_{lt}^l(r_{lt})$ is concave in $r_{lt}$. According to the first-order condition of $\frac{d\Pi_{lt}^l(r_{lt})}{dr_{lt}} = 0$, we get $\widetilde{r_{lt}} = \frac{((\omega_{lt}+t_{lt})q_{lt}^*(r_{lt})-b)(p-\varepsilon)f(q_{lt}^*(r_{lt}))-(\omega_{lt}+t_{lt})(t_{lt}-n)}{(\omega_{lt}+t_{lt})^2}$, since $r_{lt} \geq 0$, we can obtain

$$r_{lt}^* = \begin{cases} \widetilde{r_{lt}}, 0 \leq b < \frac{(\omega_{lt}+t_{lt})q_{lt}^*(r_{lt})(p-\varepsilon)f(q_{lt}^*(r_{lt}))-(\omega_{lt}+t_{lt})(t_{lt}-n)}{(p-\varepsilon)f(q_{lt}^*(r_{lt}))} \\ 0, b \geq \frac{(\omega_{lt}+t_{lt})q_{lt}^*(r_{lt})(p-\varepsilon)f(q_{lt}^*(r_{lt}))-(\omega_{lt}+t_{lt})(t_{lt}-n)}{(p-\varepsilon)f(q_{lt}^*(r_{lt}))} \end{cases}.$$

(3) According to Equation (7), similar to the analysis in the Proof of Proposition 1, since the inequality $\frac{(1+r_{lt})(\omega_{lt}-c-p_c e_1)}{2(p-\varepsilon)} < \frac{(\omega_{lt}+t_{lt})(1+r_{lt})+p_c e_0-\varepsilon}{p-\varepsilon} = \overline{F}(q_{lt}^*(\omega_{lt}))$ holds, we can get $\frac{d^2\Pi_{lt}^s(\omega_{lt})}{d\omega_{lt}^2} = \frac{dq_{lt}^*(\omega_{lt})}{d\omega_{lt}}\left(\frac{2(p-\varepsilon)f^2(q_{lt}^*(\omega_{lt}))+(1+r_{lt})(\omega_{lt}-c-p_c e_1)f'(q_{lt}^*(\omega_{lt}))}{f^2(q_{lt}^*(\omega_{lt}))(p-\varepsilon)}\right) < 0$. Thus, we know that $\Pi_{lt}^s(\omega_{lt})$ is concave in $\omega_{lt}$. According to the first-order condition of $\frac{d\Pi_{lt}^s(\omega_{lt})}{d\omega_{lt}} = 0$, we can get $\omega_{lt}^* = \frac{(1+r_{lt}^*(\omega_{lt}^*))(H(q_{lt}^*(\omega_{lt}^*))t_{lt}+c+p_c e_1)+(p_c e_0-\varepsilon)H(q_{lt}^*(\omega_{lt}^*))}{(1-H(q_{lt}^*(\omega_{lt}^*)))(1+r_{lt}^*(\omega_{lt}^*))}$. □

**Proof of Proposition 4.** (1) According to Equation (8), similar to the analysis in the Proof of Proposition 1, we can get $\frac{d\Pi_{tl}^m(q_{tl})}{dq_{tl}} = (p - \varepsilon)\overline{F}(q_{tl}) + \varepsilon - p_c e_0 - \omega_{tl} - t_{tl}$, $\frac{d^2\Pi_{tl}^m(q_{tl})}{dq_{tl}^2} = -(p - \varepsilon)f(q_{tl}) < 0$. Therefore, let $\frac{d\Pi_{tl}^m(q_{tl})}{dq_{tl}} = 0$, we can get $q_{tl}^* = \overline{F}^{-1}\left(\frac{\omega_{tl}+t_{tl}+p_c e_0-\varepsilon}{p-\varepsilon}\right)$.

(2) According to Equation (9), since the inequality $\frac{\omega_{tl}-c(1+r_{tl})-p_c e_1}{2(p-\varepsilon)} < \frac{\omega_{tl}+t_{tl}+p_c e_0-\varepsilon}{p-\varepsilon} = \overline{F}(q_{tl}^*(\omega_{tl}))$ holds, we can get $\frac{d^2\Pi_{tl}^s(\omega_{tl})}{d\omega_{tl}^2} = \frac{dq_{tl}^*(\omega_{tl})}{d\omega_{tl}}\left(\frac{2(p-\varepsilon)f^2(q_{tl}^*(\omega_{tl}))+(\omega_{tl}-c(1+r_{tl})-p_c e_1)f'(q_{tl}^*(\omega_{tl}))}{f^2(q_{tl}^*(\omega_{tl}))(p-\varepsilon)}\right) < 0$. Thus, $\Pi_{tl}^s(\omega_{tl})$ is concave in $\omega_{tl}$. Let $\frac{d\Pi_{tl}^s(\omega_{tl})}{d\omega_{tl}} = 0$, we obtain $\omega_{tl}^* = \frac{H(q_{tl}^*(\omega_{tl}^*))(t_{tl}+p_c e_0-\varepsilon)+c(1+r_{tl})+p_c e_1}{1-H(q_{tl}^*(\omega_{tl}^*))}$.

(3) According to Equation (10), we can get $\frac{d\Pi_{tl}^s(r_{tl})}{dr_{tl}} = cq_{tl}^*(r_{tl}) - b + (cr_{tl} + t_{tl} + n)\frac{dq_{tl}^*(r_{tl})}{dr_{tl}}$, $\frac{d^2\Pi_{tl}^s(r_{tl})}{dr_{tl}^2} = 2c\frac{dq_{tl}^*(r_{tl})}{dr_{tl}} + (cr_{tl} + t_{tl} - n)\frac{d^2q_{tl}^*(r_{tl})}{dr_{tl}^2}$. Based on the analysis of Proposition 4, we can get $\frac{dq_{tl}^*(r_{tl})}{dr_{tl}} = \frac{dq_{tl}^*(r_{tl})}{d\omega_{tl}^*(r_{tl})}\frac{d\omega_{tl}^*(r_{tl})}{dr_{tl}}$, $\frac{d\omega_{tl}^*(r_{tl})}{dr_{tl}} = c$. Thus, it is obvious that $\Pi_{tl}^l(r_{tl})$ is concave in $r_{tl}$. Let $\frac{d\Pi_{tl}^l(r_{tl})}{dr_{tl}} = 0$, we can get $r_{tl}^* = \frac{c(n-t_{tl})-f(q_{tl}^*(r_{tl}))(p-\varepsilon)(b-cq_{tl}^*(r_{tl}))}{c^2}$. □

**Proof of Proposition 5.** (1) According to Equation (11), we can get $\frac{d\Pi_{tt}^m(q_{tt})}{dq_{tt}} = (p - \varepsilon)\overline{F}(q_{tt}) + \varepsilon - p_c e_0 - (\omega_{tt} + t_{tt})(1 + r_m)$, $\frac{d^2\Pi_{tt}^m(q_{tt})}{dq_{tt}^2} = -(p - \varepsilon)f(q_{tt}) < 0$. As a result, we know that $\Pi_{tt}^m(q_{tt})$ is concave in $q_{tt}$. According to the first-order condition of $\frac{d\Pi_{tt}^m(q_{tt})}{dq_{tt}} = 0$, we get $q_{tt}^* = \overline{F}^{-1}\left(\frac{(\omega_{tt}+t_{tt})(1+r_m)+p_c e_0-\varepsilon}{p-\varepsilon}\right)$.

(2) According to Equation (12), similar to the analysis in the Proof of Proposition 3 and 4, since the inequality $\frac{(1+r_m)(\omega_{tt}-c(1+r_s)-p_c e_1)}{2(p-\varepsilon)} < \frac{(\omega_{tt}+t_{tt})(1+r_m)+p_c e_0-\varepsilon}{p-\varepsilon} = \overline{F}(q_{tt}^*(\omega_{tt}))$ holds, we get $\frac{d^2\Pi_{tt}^s(\omega_{tt})}{d\omega_{tt}^2} = \frac{dq_{tt}^*(\omega_{tt})}{d\omega_{tt}}\left(\frac{2(p-\varepsilon)f^2(q_{tt}^*(\omega_{tt}))+(1+r_m)(\omega_{tt}-c(1+r_s)-p_c e_1)f'(q_{tt}^*(\omega_{tt}))}{f^2(q_{tt}^*(\omega_{tt}))(p-\varepsilon)}\right) < 0$. Thus, we know that $\Pi_{tt}^s(\omega_{tt})$ is concave in $\omega_{tt}$. According to the first-order condition of $\frac{d\Pi_{tt}^s(\omega_{tt})}{d\omega_{tt}} = 0$, we get $\omega_{tt}^* = \frac{(1+r_m)(H(q_{tt}^*(\omega_{tt}^*))t_{tt}+c(1+r_s)+p_c e_1)+(p_c e_0-\varepsilon)H(q_t^*(\omega_{tt}^*)))}{(1-H(q_{tt}^*(\omega_{tt}^*)))(1+r_m)}$.

(3) According to Equation (14), similar to the analysis in the Proof of Proposition 3 and 4, we can get $\frac{d^2\Pi_{tm}^s(r_m)}{dr_m^2} = (\omega_{tt} + t_{tt})\frac{dq_{tt}^*(r_m)}{dr_m}\left(\frac{2(p-\varepsilon)f^2(q_{tt}^*(r_m))+((\omega_{tt}+t_{tt})r_m+t_{tt}-n)f'(q_{tt}^*(r_m))}{f^2(q_{tt}^*(r_m))(p-\varepsilon)}\right)$,

(i) If $f'(q_{tt}^*(r_m)) \geq 0$, then we have $\frac{d^2\Pi_{tt}^s(r_m)}{dr_{tt}^2} < 0$,

(ii) If $f'(q_{tt}^*(\omega_{tt})) < 0$, since $\frac{(\omega_{tt}+t_{tt})r_m+t_{tt}-n}{2(p-\varepsilon)} < \frac{(\omega_{tt}+t_{tt})(1+r_m)+p_c e_0-\varepsilon}{p-\varepsilon} = \overline{F}(q_{tt}^*(r_m))$, thus,

$$\frac{d^2\Pi_{tt}^s(r_m)}{dr_{tt}^2} = (\omega_{tt} + t_{tt})\frac{dq_{tt}^*(r_m)}{dr_{tt}}\left(\frac{2(p-\varepsilon)f^2(q_{tt}^*(r_{tt}))+((\omega_{tt}+t_{tt})r_m+t_{tt}-n)f'(q_{tt}^*(r_{tt}))}{f^2(q_{tt}^*(r_{tt}))(p-\varepsilon)}\right)$$
$$< 2(\omega_{tt} + t_{tt})\frac{dq_{tt}^*(r_m)}{dr_{tt}}\left(\frac{f^2(q_{tt}^*(r_m))+\overline{F}(q_{tt}^*(r_m))f'(q_{tt}^*(r_m))}{f^2(q_{tt}^*(r_m))}\right)$$
$$= 2(\omega_{tt} + t_{tt})\frac{dq_{tt}^*(r_m)}{dr_{tt}}\left(\frac{h'(q_{tt}^*(r_m))\overline{F}(q_{tt}^*(r_m))}{f^2(q_{tt}^*(r_m))}\right) < 0$$

We know that $\Pi_{tt}^l(r_m)$ is concave in $r_m$. According to the first-order condition of $\frac{d\Pi_{tt}^l(r_m)}{dr_m} = 0$, we get $\widetilde{r_m} = \frac{((\omega_{tt}^*(r_m)+t_{tt})q_{tt}^*(r_m)-b)(p-\varepsilon)f(q_{tt}^*(r_m))-(\omega_{tt}^*(r_m)+t_{tt})(t_{tt}-n)}{(\omega_{tt}^*(r_m)+t_{tt})^2}$, since $r_m \geq 0$, we obtain $r_m^* = \begin{cases} \widetilde{r_m}, 0 \leq b < \frac{((\omega_{tt}^*(r_m)+t_{tt})q_{tt}^*(r_m)-b)(p-\varepsilon)f(q_{tt}^*(r_m))-(\omega_{tt}^*(r_m)+t_{tt})(t_{tt}-n)}{(\omega_{tt}^*(r_m)+t_{tt})^2} \\ 0, b \geq \frac{((\omega_{tt}^*(r_m)+t_{tt})q_{tt}^*(r_m)-b)(p-\varepsilon)f(q_{tt}^*(r_m))-(\omega_{tt}^*(r_m)+t_{tt})(t_{tt}-n)}{(\omega_{tt}^*(r_m)+t_{tt})^2} \end{cases}$.

(4) According to Equation (15), we can get $\frac{d\Pi_{ts}^s(r_s)}{dr_s} = cq_{tt}^*(r_s) - s + cr_s\frac{dq_{tt}^*(r_s)}{dr_s}$, $\frac{d^2\Pi_{ts}^s(r_s)}{dr_s^2} = 2c\frac{dq_{tt}^*(r_s)}{dr_s} + cr_s\frac{d^2q_{tt}^*(r_s)}{dr_s^2}$. Based on the analysis of Proposition 5, we know that $\frac{dq_{tt}^*(r_s)}{dr_s} = \frac{dq_{tt}^*(r_s)}{d\omega_{tt}^*(r_s)}\frac{d\omega_{tt}^*(r_s)}{dr_s}$, $\frac{d\omega_{tt}^*(r_s)}{dr_s} = c$. Thus, $\Pi_{ts}^l(r_s)$ is concave in $r_s$. Let $\frac{d\Pi_{ts}^l(r_s)}{dr_s} = 0$, we can obtain $r_s^* = \frac{f(q_{tt}^*(r_s))(p-\varepsilon)(cq_{tt}^*(r_s)-s)}{c^2(1+r_m)}$. □

**Proof of Proposition 6.** According to Equation (25), taking the second-order derivatives of $U(\Pi_{sa}^m(q_{sa}))$ with respect to $q_{sa}$, we get $\frac{d^2U(\Pi_{sa}^m(q_{sa}))}{dq_{sa}^2} = -[(p-\varepsilon)f(q_{sa}) + \frac{(\lambda-1)((\omega_{sa}+t_{sa})(1+r_{sa})+p_c e_0-\varepsilon)^2}{(p-\varepsilon)}f(x_0)] < 0$. Therefore, let $\frac{dU(\Pi_{sa}^m(q_{sa}))}{dq_{sa}} = 0$, we can get $q_{sa}^* = \overline{F}^{-1}\left(\frac{(\omega_{sa}+t_{sa}+p_c e_0-\varepsilon)(\lambda-(\lambda-1)\overline{F}(x_0))}{(p-\varepsilon)}\right)$.

According to Equation (24), we can obtain the optimal loan interest rate $r_{sa}^* = 0$ and $\frac{d\Pi_{sa}^s(\omega_{sa})}{d\omega_{sa}} = q_{sa}^*(\omega_{sa})(1 - F(x_0)) + (\omega_{sa} - c - p_c e_1 - (\omega_{sa} + t_{sa} - p_c e_0 - \varepsilon)F(x_0))(dq_{sa}^*(\omega_{sa})/d\omega_{sa})$. Let $\frac{d\Pi_{sa}^s(\omega_{sa})}{d\omega_{sa}} = 0$, we get $\omega_{sa}^* = \frac{F(x_0)(t_{sa}-p_c e_0-\varepsilon)+c+p_c e_1-\overline{F}(x_0)q_{sa}^*(\omega_{sa})(d\omega_{sa}/dq_{sa}^*(\omega_{sa}))}{\overline{F}(x_0)}$, where, $\frac{dq_{sa}^*}{d\omega_{sa}} = \frac{-((\lambda-1)((\omega_{sa}+t_{sa}+p_c e_0-\varepsilon)q_{sa}^*(\omega_{sa})f(x_0))+(p-\varepsilon)(\lambda-(\lambda-1)\overline{F}(x_0)))}{(p-\varepsilon)^2f(q_{sa}^*(\omega_{sa}))+(\lambda-1)(\omega_{sa}+t_{sa}+p_c e_0-\varepsilon)^2f(x_0)}$. □

**Proof of Proposition 7.** Similar to the work of Proposition 6, according to Equation (26), we can get $\frac{d^2U(\Pi_{la}^m(q_{la}))}{dq_{la}^2} < 0$. Let $\frac{dU(\Pi_{la}^m(q_{la}))}{dq_{la}} = 0$, we can get $q_{la}^* = \overline{F}^{-1}\left(\frac{((\omega_{la}+t_{la})(1+r_{la})+p_c e_0-\varepsilon)(\lambda-(\lambda-1)\overline{F}(x_1))}{(p-\varepsilon)}\right)$, where, $x_1 = \frac{((\omega_{la}+t_{la})q_{la}-b)(1+r_{la})+b-p_c(M-e_0q_{la})-\varepsilon q_{la}}{p-\varepsilon}$. And from Equation (27), we know that $\frac{d\Pi_{la}^l(r_{la})}{dr_{la}} = ((\omega_{la} + t_{la})r_{la} + t_{la} - n)\frac{dq_{la}^*(r_{la})}{dr_{la}} - (p - \varepsilon)F(x_1)\frac{dx_1}{dr_{la}} +$

$(\omega_{la} + t_{la})q_{la}^* - b$, $\dfrac{d^2\Pi_{la}^l(r_{la})}{dr_{la}^2} = 2(\omega_{la} + t_{la})\dfrac{dq_{la}^*(r_{la})}{dr_{la}} + ((\omega_{la} + t_{la})r_{la} + t_{la} - n)\dfrac{d^2q_{la}^*(r_{la})}{dr_{la}^2}$
$-(p - \varepsilon)f(x_1)(\dfrac{dx_1(r_{la})}{dr_{la}})^2 - (p - \varepsilon)F(x_1)\dfrac{d^2x_1}{dr_{la}^2} < 0$ . Therefore, from

$\dfrac{d\Pi_{la}^l(r_{la})}{dr_{la}} = 0$, we get $r_{la}^* = \dfrac{F(x_1)(\omega_{la} + t_{la} - p_c e_0 - \varepsilon) + n - t_{la} - \overline{F}(x_1)((\omega_{la} + t_{la})q_{la}^*(r_{la}) - b)(dr_{la}/dq_{la}^*(r_{la}))}{(\omega_{la} + t_{la})\overline{F}(x_1)}$, where,

$\dfrac{dq_{la}^*}{dr_{la}} = \dfrac{-(\omega_{la} + t_{la})((\lambda - 1)((\omega_{la} + t_{la})(1 + r_{la}) + p_c e_0 - \varepsilon)q_{la}^*(r_{la})f(x_1) + (p - \varepsilon)(\lambda - (\lambda - 1)\overline{F}(x_1))) - b(\lambda - 1)((\omega_{la} + t_{la})(1 + r_{la}) + p_c e_0 - \varepsilon)f(x_1)}{(p - \varepsilon)^2 f(q_{la}^*(r_{la})) + (\lambda - 1)((\omega_{la} + t_{la})(1 + r_{la}) + p_c e_0 - \varepsilon)^2 f(x_1)}$,

According to Equation (28), we can get $\dfrac{d\Pi_{la}^s(\omega_{la})}{d\omega_{la}} = q_{la}^*(\omega_{la}) + (\omega_{la} - c - p_c e_1)\dfrac{dq_{la}^*(\omega_{la})}{d\omega_{la}}$,

$\dfrac{d^2\Pi_{la}^s(\omega_{la})}{d\omega_{la}^2} = 2\dfrac{dq_{la}^*(\omega_{la})}{d\omega_{la}} + (\omega_{la} - c - p_c e_1)\dfrac{d^2q_{la}^*(\omega_{la})}{d\omega_{la}^2}$. Therefore, according to the first-order

condition of $\dfrac{d\Pi_{la}^s(\omega_{la})}{d\omega_{la}} = 0$, we get $\omega_{la}^* = c + p_c e_1 - \dfrac{q_{la}^*(\omega_{la})}{(dq_{la}^*(\omega_{la})/d\omega_{la})}$, where, $\dfrac{dq_{la}^*}{d\omega_{la}} =$

$\dfrac{-(1 + r_{la}^*(\omega_{la}))((\lambda - 1)((\omega_{la} + t_{la})(1 + r_{la}^*(\omega_{la})) + p_c e_0 - \varepsilon)q_{la}^*(\omega_{la})f(x_1) + (p - \varepsilon)(\lambda - (\lambda - 1)\overline{F}(x_1)))}{(p - \varepsilon)^2 f(q_{la}^*(\omega_{la})) + (\lambda - 1)((\omega_{la} + t_{la})(1 + r_{la}^*(\omega_{la})) + p_c e_0 - \varepsilon)^2 f(x_1)}$. $\square$

## Appendix C. Proof of Corollary

**Proof of Corollary 1.** According to Proposition 1–5, taking the first-order of $q_j^*$ with respect to $\omega$, $r$, $t$, $p_c$, $e_0$, $e_1$ and $\varepsilon$, respectively, where, $j = nn, st, lt, tl, tt$. we can get $\dfrac{dq_j^*(\omega_j)}{d\omega_j} = \dfrac{dq_j^*(t_j)}{dt_j} = \dfrac{-(1 + r_j)}{f(q_j^*)(p - \varepsilon)} < 0$,

$\dfrac{dq_j^*(r_j)}{dr_j} = \dfrac{-(\omega_j + r_j)}{f(q_j^*(r_j))(p - \varepsilon)} < 0$, $\dfrac{dq_j^*(p_c)}{dp_c} = \dfrac{-e_0}{f(q_j^*(p_c))(p - \varepsilon)} < 0$, $\dfrac{dq_j^*(e_0)}{de_0} = \dfrac{-p_c}{f(q_j^*(e_0))(p - \varepsilon)} < 0$, $\dfrac{dq_j^*(e_0)}{de_0} = \dfrac{-p_c}{f(q_j^*(e_0))(p - \varepsilon)} <$

$0$, $\dfrac{dq_j^*(\varepsilon)}{d\varepsilon} = \dfrac{p - \varepsilon + (\omega_j + t_j)(1 + r_j) + p_c e_0 - \varepsilon}{f(q_j^*(\varepsilon))(p - \varepsilon)^2} > 0$. $\square$

**Proof of Corollary 2.** (1) According to Proposition 1–5, taking the first-order of $\omega_{j2}^*$ with respect to $p_c$, $e_0$, $e_1$ and $\varepsilon$, respectively, where, $j2 = nn, st, lt, tt$, $j3 = lt, m$, we can get $\dfrac{d\omega_{j2}^*(p_c)}{dp_c} =$

$\dfrac{\partial\omega_{j2}^*(p_c)}{\partial q_{j2}^*(p_c)}\dfrac{\partial q_{j2}^*(p_c)}{\partial p_c} + \dfrac{e_0 H(q_{j2}^*(p_c)) + e_1(1 + r_{j3}^*(p_c))}{(1 - H(q_{j2}^*(p_c)))(1 + r_{j3}^*(p_c))} > 0$, $\dfrac{d\omega_{j2}^*(e_0)}{de_0} = \dfrac{\partial\omega_{j2}^*(e_0)}{\partial q_{j2}^*(e_0)}\dfrac{\partial q_{j2}^*(e_0)}{\partial e_0} + \dfrac{p_c H(q_{j2}^*(e_0))}{(1 - H(q_{j2}^*(e_0)))(1 + r_{j3}^*(e_0))} > 0$,

$\dfrac{d\omega_{j2}^*(e_1)}{de_1} = \dfrac{p_c(1 + r_{j3}^*(e_1))}{1 - H(q_{j2}^*(e_1))} > 0$, $\dfrac{d\omega_{j2}^*(\varepsilon)}{d\varepsilon} = \dfrac{\partial\omega_{j2}^*(\varepsilon)}{\partial q_{j2}^*(\varepsilon)}\dfrac{\partial q_{j2}^*(\varepsilon)}{\partial\varepsilon} - \dfrac{H(q_{j2}^*(\varepsilon))}{(1 - H(q_{j2}^*(\varepsilon)))(1 + r_{j3}^*(\varepsilon))} < 0$,

(2) According to Proposition 4, taking the first-order of $\omega_{tl}^*$ with respect to $p_c$, $e_0$, $e_1$ and $\varepsilon$, respectively, we can get $\dfrac{d\omega_{tl}^*(p_c)}{dp_c} = \dfrac{\partial\omega_{tl}^*(p_c)}{\partial q_{tl}^*(p_c)}\dfrac{\partial q_{tl}^*(p_c)}{\partial p_c} + \dfrac{e_0 H(q_{tl}^*(p_c)) + e_1}{1 - H(q_{tl}^*(p_c))} > 0$, $\dfrac{d\omega_{tl}^*(e_0)}{de_0} = \dfrac{\partial\omega_{tl}^*(e_0)}{\partial q_{tl}^*(e_0)}\dfrac{\partial q_{tl}^*(e_0)}{\partial e_0} + \dfrac{p_c H(q_{tl}^*(e_0))}{1 - H(q_{tl}^*(e_0))} > 0$,

$\dfrac{d\omega_{tl}^*(e_1)}{de_1} = \dfrac{p_c H(q_{tl}^*(e_1))}{1 - H(q_{tl}^*(e_1))} > 0$, $\dfrac{d\omega_{tl}^*(\varepsilon)}{d\varepsilon} = \dfrac{\partial\omega_{tl}^*(\varepsilon)}{\partial q_{tl}^*(\varepsilon)}\dfrac{\partial q_{tl}^*(\varepsilon)}{\partial\varepsilon} - \dfrac{H(q_{tl}^*(\varepsilon))}{1 - H(q_{tl}^*(\varepsilon))} < 0$. $\square$

**Proof of Corollary 3.** (1) According to Proposition 3 and 5, we can observe when $0 \le b < b_1$, let $b = (\omega_{j2} + t_{j2})q_{j2}^*(r_{j3}) - \dfrac{(\omega_{j2} + t_{j2})(t_{j2} - n) - b_1}{(p - \varepsilon)f(q_{j2}^*(r_{j3}))}$, we can get $r_{j3}^* = \dfrac{b_1}{(\omega_{j2} + t_{j2})^2}$, where, $0 < b_1 \le (\omega_{j2} + t_{j2})q_{j2}^*(r_{j3})(p - \varepsilon)f(q_{j2}^*(r_{j3})) - (\omega_{j2} + t_{j2})(t_{j2} - n)$. Taking the first-order of $r_{j3}^*$ with respect to $\omega_{j2}$, $t_{j2}$ respectively,

we can get $\dfrac{dr_{j3}^*(\omega_{j2})}{d\omega_{j2}} = \dfrac{\partial r_{j3}^*(\omega_{j2})}{\partial q_{j2}^*(\omega_{j2})}\dfrac{\partial q_{j2}^*(\omega_{j2})}{\partial\omega_{j2}} + \dfrac{\partial r_{j3}^*(\omega_{j2})}{\partial\omega_{j2}}$, $\dfrac{dr_{j3}^*(t_{j2})}{dt_{j2}} = \dfrac{\partial r_{j3}^*(t_{j2})}{\partial q_{j2}^*(t_{j2})}\dfrac{\partial q_{j2}^*(t_{j2})}{\partial t_{j2}} + \dfrac{\partial r_{j3}^*(t_{j2})}{\partial t_{j2}}$, thus, we know

$\dfrac{dr_{j3}^*(\omega_{j2})}{d\omega_{j2}} = \dfrac{dr_{j3}^*(t_{j2})}{dt_{j2}} = \dfrac{-2b_1}{(\omega_{j2} + t_{j2})^3} < 0$, When $b \ge b_1$, we have $r_{j3}^* = 0$, so we can get $\dfrac{dr_{j3}^*(\omega_{j2})}{d\omega_{j2}} = \dfrac{dr_{j3}^*(t_{j2})}{dt_{j2}} = 0$.

Therefore, the inequality of $\dfrac{dr_{j3}^*(\omega_{j2})}{d\omega_{j2}} \le 0$, $\dfrac{dr_{j3}^*(t_{j2})}{dt_{j2}} \le 0$ holds. According to Proposition 3 and 5, taking the first-order of $r_{j3}^*$ with respect to $p$, $\varepsilon$ and $b$ respectively. We find that when $0 \le b < b_1$, we can get

$\dfrac{dr_{j3}^*(p)}{dp} = \dfrac{\partial r_{j3}^*(p)}{\partial q_{j2}^*(p)}\dfrac{\partial q_{j2}^*(p)}{\partial p} + \dfrac{((\omega_{j2} + t_{j2})q_{j2}^*(p) - b)f(q_{j2}^*(p))}{(\omega_{j2} + t_{j2})^2} > 0$, $\dfrac{dr_{j3}^*(\varepsilon)}{d\varepsilon} = \dfrac{\partial r_{j3}^*(\varepsilon)}{\partial q_{j2}^*(\varepsilon)}\dfrac{\partial q_{j2}^*(\varepsilon)}{\partial\varepsilon} - \dfrac{((\omega_{j2} + t_{j2})q_{j2}^*(\varepsilon) - b)f(q_{j2}^*(\varepsilon))}{(\omega_{j2} + t_{j2})^2} < 0$,

$\dfrac{dr_{j3}^*(b)}{db} = \dfrac{-(p - \varepsilon)f(q_{j2}^*(b))}{(\omega_{j2} + t_{j2})^2} < 0$, and when $b \ge b_1$, we can get $\dfrac{dr_{j3}^*(p)}{dp} = 0$, $\dfrac{dr_{j3}^*(\varepsilon)}{d\varepsilon} = 0$, $\dfrac{dr_{j3}^*(b)}{db} = 0$, where,

$b_1 = \frac{(\omega_{j2}+t_{j2})q^*_{j2}(r_{j3})(p-\varepsilon)f(q^*_{j2}(r_{j3}))-(\omega_{j2}+t_{j2})(t_{j2}-n)}{(p-\varepsilon)f(q^*_{j2}(r_{j3}))}$. As a result, we know that the inequality of $\frac{dr^*_{j3}(p)}{dp} \geq 0$, $\frac{dr^*_{j3}(\varepsilon)}{d\varepsilon} \leq 0$, $\frac{dr^*_{j3}(b)}{db} \leq 0$ holds.

(2) According to Proposition 4 and 5, taking the first-order of $r^*_{j4}$ with respect to $p$, $t$, $\varepsilon$ and $s$, respectively, we can get $\frac{dr^*_{j5}(p)}{dp} = \frac{\partial r^*_{j5}(p)}{\partial q^*_{j4}(p)}\frac{\partial q^*_{j4}(p)}{\partial p} + \frac{f(q^*_{j4}(p))(cq^*_{j4}(p)-b)}{c^2} \geq 0$, $\frac{dr^*_{j5}(t_{j4})}{dt_{j4}} = \frac{\partial r^*_{j5}(t_{j4})}{\partial q^*_{j4}(t_{j4})}\frac{\partial q^*_{j4}(t_{j4})}{\partial t_{j4}} - \frac{1}{c} \leq 0$, $\frac{dr^*_{j5}(s)}{ds} = \frac{\partial r^*_{j5}(s)}{\partial q^*_{j4}(s)}\frac{\partial q^*_{j4}(s)}{\partial s} - \frac{f(q^*_{j4}(s)(p-\varepsilon)}{c^2} \leq 0$, $\frac{dr^*_{j5}(\varepsilon)}{d\varepsilon} = \frac{\partial r^*_{j5}(\varepsilon)}{\partial q^*_{j4}(\varepsilon)}\frac{\partial q^*_{j4}(\varepsilon)}{\partial \varepsilon} + \frac{f(q^*_{j4}(\varepsilon))(b-cq^*_{j4}(\varepsilon))}{c^2} \leq 0$, where, $j4 = tl, tt$, $j5 = tl, s$. □

**Proof of Corollary 4.** (1) We assume that 3PL firm adopts fixed transportation fee strategy, that is $t_j = t$, then put $\omega^*_{nn}$ into $q^*_{nn}$, we can get $\overline{F}(q^*_{nn})(1 - H(q^*_{nn})) = \frac{c-\varepsilon+t+p_c(e_0+e_1)}{p-\varepsilon}$. Similarly, we can get $\overline{F}(q^*_{st})(1 - H(q^*_{st})) = \frac{c-\varepsilon+t+p_c(e_0+e_1)}{p-\varepsilon}$, $\overline{F}(q^*_{lt})(1 - H(q^*_{lt})) = \frac{(c+t)(1+r_{lt})+p_c(e_0+e_1(1+r_{lt}))-\varepsilon}{p-\varepsilon}$, $\overline{F}(q^*_{tl})(1 - H(q^*_{tl})) = \frac{(c(1+r_s)+t)(1+r_m)+p_c(e_0+e_1(1+r_m))-\varepsilon}{p-\varepsilon}$. And since $\overline{F}(q)(1 - H(q))$ is a monotonically decreasing functions of $q$, we can know that $q^*_{st} = q^*_{nn} \geq q^*_{j6}$ holds.

(2) For the optimal strategies of $\omega^*_{lt}$ and $\omega^*_{st}$, we can get $\omega^*_{lt} \leq \omega^*_{st} = \omega^*_{nn}$. We also have $\omega^*_{tl} = \frac{f(q^*_{tl})(t+p_c e_0-p)+H(q^*_{tl})(t+p_c e_0-\varepsilon)+c+p_c e_1}{1-f(q^*_{tl})-H(q^*_{tl})}$ by putting $r^*_{tl}$ into $\omega^*_{tl}$, and compare $\omega^*_{tl}$ with $\omega^*_{st}$, we get if $p \leq t + p_c e_0 - \frac{H(q^*_{st})(t+p_c e_0-\varepsilon)}{(1-H(q^*_{st}))f(q^*_{tl})} + \frac{H(q^*_{tl})(t+p_c e_0-\varepsilon)}{(1-H(q^*_{tl}))f(q^*_{tl})}$, we have $\omega^*_{tl} \geq \omega^*_{st} = \omega^*_{nn}$, otherwise, $\omega^*_{tl} \leq \omega^*_{st} = \omega^*_{nn}$.

(3) We know that if $r^*_{j6} = 0$, where, $j6 = lt, tl, tt$, it can motivate the value of $\Pi^m_{j6}(q^*_{j6})$ to the maximum. From the Equations (5), (8) and (11), we can get $\Pi^m_{j6\max}(q^*_{j6}) = \Pi^m_{st}(q^*_{j6}) = \Pi^m_{nn}(q^*_{j6})$. Since the inequality $0 < q \leq q^*_{st} = q^*_{nn}$ holds and $\Pi^m(q)$ increases with $q$, we can also get the inequality of $\Pi^m_{j6}(q^*_{j6}) \leq \Pi^m_{j6\max}(q^*_{j6}) \leq \Pi^m_{nn}(q^*_{nn}) = \Pi^m_{st}(q^*_{st})$ holds. □

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
