# Peer review of "Research on Operation and Financing Strategy of an Emission-Dependent Supply Chain under Variable Transportation Fee Strategy"

_sustainability, doi:10.3390/su11164363_

Round 1
Reviewer 1 Report
The article is interesting and well written.
In my opinion, the numbering of the figures should be reviewed, since on page 19 it repeats the numbering of the figures starting again at Figure 1.
The figures from 2 to 6 should also be improved, in the current state they cannot be read properly.
In my opinion, it would be convenient to expand the table 1, including all the symbols and notations for sub-scripts and supper-scripts.
Although it is necessary, from section 4 onwards there are a large number of equations, and probably the reader may lose interest in the article. Check if you can pass part of these formulas to the annex, of the same way that has been done with Appendix A-B.
Author Response
August 6, 2019
Dear Reviewers:
I quite appreciate your insightful comments. Those comments are all valuable and very helpful for revising and improving our paper, as well as the important guiding significance to our researches. We have studied comments carefully and have revised the Sustainability-561596 exactly. We hope this revision can make our paper more acceptable. Revised portion are marked in the paper, where the blue parts represent the deleted parts and the red parts represent the added parts. The revision were addressed point and point below.
Response to comment: The numbering of the figures should be reviewed, since on page 19 it repeats the numbering of the figures starting again at Figure 1. The figures from 2 to 6 should also be improved, in the current state they cannot be read properly.
Response: Thank you for your valuable comments. We are very sorry for our negligence of the numbering of the figures. We have made correction according to the reviewer's comments, that is, the numbering of the figures have been adjusted again. The specific corrections can be seen in the highlighted section on page 17-25 of the article.
Response to comment: It would be convenient to expand the table 1, including all the symbols and notations for sub-scripts and super-scripts.
Response: Thank you for your valuable comments. It is really true as reviewer suggested that the extension of Table 1 will make it easier for the reader to understand all the symbols and notations for sub-scripts and super-scripts. According to the reviewer's suggestion, we have re-written this part and added explanations for all the symbols and notations for sub-scripts and super-scripts. The specific corrections can be seen in the highlighted section on page 7-8 of the article.
Response to comment: Although it is necessary, from section 4 onwards there are a large number of equations, and probably the reader may lose interest in the article. Check if you can pass part of these formulas to the annex, of the same way that has been done with Appendix A-B.
Response: Thank you for your valuable comments. We are very sorry for our negligence of the number of equations in the article, which is inconvenient for readers to read. Considering the reviewer's suggestion, we have added an appendix A to summarize proposition 1-7, thus reducing the number of equations in Parts 4 and 5. The structure of the article becomes clear for the reader to read. The specific corrections can be seen in the highlighted section on page 27-30 of the article.
Special thanks to you for your good comments.
Thank you and best regards.
Yours sincerely,
Chunxia Li

Reviewer 2 Report
Comments on sustainability-561596
In this manuscript, C. Li et al. calculated and analyzed in detail for the 3PL financing service mode in an emission-dependent supply chain under the cap-and-trade system. The manuscript is well organized and the calculated data is analyzed properly. Therefore, I recommend publishing this work in Sustainability.
One minor suggestion is to summarize the conclusions with shortening the length.

Author Response
August 6, 2019
Dear Reviewers:
I quite appreciate your insightful comments. Those comments are all valuable and very helpful for revising and improving our paper, as well as the important guiding significance to our researches. We have studied comments carefully and have revised the Sustainability-561596 exactly. We hope this revision can make our paper more acceptable. Revised portion are marked in the paper, where the blue parts represent the deleted parts and the red parts represent the added parts. The revision were addressed point and point below.
Response to comment: I recommend publishing this work in Sustainability. One minor suggestion is to summarize the conclusions with shortening the length.
Response: Thank you for your valuable comments. We are sorry for the inconvenience caused by the lack of conciseness and fluency in the conclusions. We have made correction according to the reviewer's comments. We re-summarized the conclusions of the article and shortened the length of the conclusion, which is convenient for understanding the article. The specific corrections can be seen in the blue and red section on page 26-27 of the article.
Special thanks to you for your good comments.
Thank you and best regards.
Yours sincerely,
Chunxia Li
